# Metallic W/WO$_2$ solid-acid catalyst boosts hydrogen evolution reaction in alkaline electrolyte

Zhigang Chen [1,2,7], Wenbin Gong[3,4,7], Juan Wang[5,7], Shuang Hou[1], Guang Yang[1], Chengfeng Zhu[1], Xiyue Fan[1], Yifan Li[1], Rui Gao[6] & Yi Cui [1]

The lack of available protons severely lowers the activity of alkaline hydrogen evolution reaction process than that in acids, which can be efficiently accelerated by tuning the coverage and chemical environment of protons on catalyst surface. However, the cycling of active sites by proton transfer is largely dependent on the utilization of noble metal catalysts because of the appealing electronic interaction between noble metal atoms and protons. Herein, an all-non-noble W/WO$_2$ metallic heterostructure serving as an efficient solid-acid catalyst exhibits remarkable hydrogen evolution reaction performance with an ultra-low overpotential of −35 mV at −10 mA/cm² and a small Tafel slope (−34 mV/dec), as well as long-term durability of hydrogen production (>50 h) at current densities of −10 and −50 mA/cm² in alkaline electrolyte. Multiple in situ and ex situ spectroscopy characterizations combining with first-principle density functional theory calculations discover that a dynamic proton-concentrated surface can be constructed on W/WO$_2$ solid-acid catalyst under ultra-low overpotentials, which enables W/WO$_2$ catalyzing alkaline hydrogen production to follow a kinetically fast Volmer-Tafel pathway with two neighboring protons recombining into a hydrogen molecule. Our strategy of solid-acid catalyst and utilization of multiple spectroscopy characterizations may provide an interesting route for designing advanced all-non-noble catalytic system towards boosting hydrogen evolution reaction performance in alkaline electrolyte.

As a clean and sustainable energy carrier, hydrogen is one of the most promising alternatives to traditional fossil fuels for addressing global energy crisis and environmental pollution[1]. One of the most economical and effective strategy of hydrogen production is electrocatalytic hydrogen evolution reaction (HER), which is driven by electricity from sustainable energies (i.e., solar and wind) without any emission of carbon dioxide, satisfying the mission of global carbon neutrality[2,3]. Alkaline HER process can avoid the acidic corrosion and dissolution issues of catalysts, and achieve high-purity hydrogen gas (>99.7%), showing an attractive and extensive application[4,5]. HER electrocatalysis usually demands the usage of noble-metal-based catalysts to lower the applied overpotentials, but the high cost and low reserve have

[1]i-lab, Vacuum Interconnected Nanotech Workstation (Nano-X), Suzhou Institute of Nano-Tech and Nano-Bionics, Chinese Academy of Sciences, Suzhou, China. [2]School of Materials Science and Engineering, Chongqing University of Technology, Chongqing, China. [3]School of Physics and Energy, Xuzhou University of Technology, Xuzhou, China. [4]Division of Nanomaterials and Jiangxi Key Lab of Carbonene Materials, Jiangxi Institute of Nanotechnology, Nanchang, China. [5]Shanghai Synchrotron Radiation Facility (SSRF), Shanghai Advanced Research Institute, Chinese Academy of Sciences, Beijing, China. [6]Department of Chemical Engineering, Waterloo Institute for Nanotechnology, Waterloo Institute for Sustainable Energy, University of Waterloo, Waterloo, ON, Canada. [7]These authors contributed equally: Zhigang Chen, Wenbin Gong, Juan Wang. ✉e-mail: ycui2015@sinano.ac.cn

restricted their widespread utilization. Moreover, compared to the direct proton-coupled electron reaction ($2H^* + 2e^- \rightarrow H_2 + *$, where $*$ represents the active site) in acidic electrolyte, it is worth noting that an additional water dissociation step ($H_2O + * + e^- \rightarrow H^* + OH^-$, also known as Volmer step) is required to produce available protons before hydrogen generation in alkaline HER process[6], while noble platinum catalyst is kinetically inefficient for the cleavage of H-OH bonds[7], usually leading to two or three orders of magnitude lower activity of alkaline HER process than that in acids[8]. Therefore, the exploration of cost-effective catalysts towards efficiently breaking H-OH bonds for proton generation is valuable and significant in alkaline HER process.

Transition metal oxides have long been advocated as highly efficient HER catalysts due to their flexible chemical and electronic structures with fascinating physical and chemical properties[9]. In particular, tungsten (W), remarkable for its complex electronic structure featuring open d and f shells[10], can form rich oxidation states ranging from +6 to 0 ($WO_{3-x}$, $0 \leq x < 3$)[11]. Tungsten oxides with substoichiometric phases have been demonstrated as potential alternatives to commercial platinum catalyst in acidic HER process[12–14], benefiting from these abundant oxygen vacancies that can afford substoichiometric $WO_{3-x}$ catalysts with favorable hydrogen adsorption energies and improved conductivity[15]. However, their alkaline HER activities have been rarely explored, because tungsten oxides featured with acidic-oxide property will be gradually dissolved in alkaline electrolyte[16–18]. Although first-row (3d) transition metals (Fe, Co, Ni) as additives have been proven to significantly improve the alkaline HER performance of tungsten oxides[19], the introduction of foreign atoms probably obscures the catalytic mechanism, since the identification of doping state of low-atomic-number 3d metals among tungsten arrays is rather challenging[20]. Therefore, it is still unclear that whether tungsten-oxide-dominant materials can be employed as stable catalysts for highly-efficient and stable alkaline HER process, when the local chemical environment of W atoms is altered.

To this end, we have focused on a special form of $W/WO_2$ metallic heterostructure. Unlike previously reported acid-like catalyst surface constructed by metal-oxide matrix supported noble-metals[18,21], where metal oxides are expected to provide an acid-like catalyst surface under neutral or alkaline conditions, while noble metals are essentially required as active sites due to the energy-favorable interactions of noble-metal and produced protons on catalyst surface[22]. The $W/WO_2$ metallic heterostructure can spontaneously construct a dynamic proton-concentrated surface only by tungsten-atom itself, thus enabling kinetically fast HER process in high-pH conditions, which can be understood in the viewpoint of solid-acid catalyst: (i) featured by unusual metallic and acidic-oxide properties, the $WO_2$ component can serve as highly-active Lewis acid sites for the adsorption and cleavage of $H_2O$ molecules[23,24] (proton generation), facilitating the formation of hydrogen tungsten bronze ($H_xWO_y$) intermediates[25] (proton storage); (ii) the in-situ generated $H_xWO_y$ intermediates are considered as Brønsted acid sites with reversible adsorption/desorption behaviors of protons[26,27]; (iii) considering the relatively sluggish hydrogen desorption kinetics of $H_xWO_y$ intermediates, the introduction of zero-valence W ($W^0$) sites can further accelerate the deprotonation kinetics of Brønsted acids for the cycling of active sites due to the optimized electronic interactions between $W^0$ atoms and protons at the $W/WO_2$ interface[28] (proton donation and regeneration of active sites). In addition, compared to traditional semiconducting tungsten oxides, the metallic feature of $WO_2$ component can also afford the tungsten-oxide matrix with improved alkaline leaching resistance ($WO_2 + OH^- \rightarrow WO_4^{2-} + H_2O$), because the aggressive hydroxyl species from produced $OH^-$ intermediates and electrolyte can be rapidly repulsed from the electron-rich catalyst surface at the cathode[29,30].

Herein, we report a feasible pyrolysis-reduction strategy to synthesize $W/WO_2$ metallic heterostructure on Ni foam. Owing to the solid-acid sites with strong abilities of proton generation and reversible behaviors of hydrogen adsorption/desorption, a dynamic proton-concentrated surface is constructed on $W/WO_2$ solid-acid catalyst, which enables the all-non-noble $W/WO_2$ catalyst to show superior HER activity with an ultra-low overpotential of −35 mV at −10 mA/cm² and a small Tafel slope (−34 mV/dec) in alkaline electrolyte. Moreover, the solid-acid catalyst is exceptionally stable in alkaline electrolyte, showing no significant activity degradation for hydrogen production at −10 and −50 mA/cm² over 50 h. To the best of our knowledge, this is the first time that a tungsten-oxide-dominant material serves as a solid-acid catalyst with a remarkable HER performance in alkaline electrolyte, outperforming all tungsten/molybdenum oxides and most 3d-metal oxides reported to date.

## Results
### Morphological characterization of $W/WO_2$ metallic heterostructure
$W/WO_2$ metallic heterostructure was synthesized by a pyrolysis-reduction method using $W_{18}O_{49}$ nanowires (NWs) as parent materials (Supplementary Fig. 1). In brief, $W_{18}O_{49}$ NWs were homogeneously coated on the pre-treated Ni foam (Supplementary Fig. 2), followed by stirring in the mixture solution of polyethylene oxide-co-polypropylene oxide-co-polyethylene oxide ($P_{123}$, carbon source), 2-amino-2-hydroxymethyl-propane-1,3-dio (Tris) and dopamine (DA, carbon source) for 24 h to coat organic carbon source over $W_{18}O_{49}$ materials. Afterwards, the desired $W/WO_2$ metallic heterostructure was obtained by pyrolyzing the free-standing organic-tungsten precursor at 700 °C under a mixture of $Ar/H_2$ (v/v = 8:1) atmosphere for 2 h (Supplementary Fig. 2), where the reduction reactions can be simplified as the equations ($C + W_{18}O_{49} \rightarrow W + WO_2 + CO_2 \uparrow$, $H_2 + W_{18}O_{49} \rightarrow W + WO_2 + H_2O \uparrow$). In the pyrolysis process, carbon can cause the reduction of parent high-valence $W_{18}O_{49}$ parent materials for the generation of desired W and $WO_2$ products. In the following alkaline HER process, the produced graphite carbon layers not only improve the electrocatalytic charge transfer, but also alleviate the alkaline-leaching rate of inner $W/WO_2$ materials. Meanwhile, the possibly formed tungsten-carbide by-products (e.g., WC, $W_2C$) are excluded by X-ray photoelectron spectroscopy (XPS) and Raman characterizations (Supplementary Fig. 3). For comparison, $WO_2$ nanorods and W NPs were also synthesized under different conditions, respectively (seen in the method section).

The crystal structure of $W/WO_2$ heterostructure was verified by X-ray diffraction (XRD) pattern. As shown in Supplementary Fig. 4, besides of the predominant characteristics of underlying Ni foam, the XRD pattern of $WO_2$ sample reveals a set of characteristic signals at 25.8°, 37.1°, 52.9°, and 59.7°, which are indexed to the (011), (−211), (220), and (031) facets of monoclinic structured $WO_2$ phase (JCPDS No. 32–1393), respectively, while metallic W NPs present a simple cubic phase with lattice parameters of a = b = c = 3.16 Å (JCPDS no. 89–2767). $W/WO_2$ metallic heterostructure shows hybrid signals containing $WO_2$ and W phases, indicating the coexistence of two types of tungsten-based phases. The overall morphology of $W/WO_2$ product was characterized by scanning electron microscopy (SEM). From low-magnification SEM image, the Ni skeletons are closely coated by $W/WO_2$ products (Supplementary Fig. 5a, b), and the high-magnification SEM image and corresponding energy dispersive X-ray spectra (EDS) show one-dimensional (1D) nanorod structure of the as-obtained $W/WO_2$ materials with radial lengths over 500 nm (Supplementary Fig. 5c, d), indicating the 1D nanostructure of tungsten-oxide precursor is still retained after high-temperature pyrolysis treatment. High-angle annular dark-field scanning transmission electron microscopy (HAADF-STEM) can allow us to observe the hybrid structure of $W/WO_2$ metallic heterostructure at atomic level. As shown in Fig. 1a, high density of nanoparticles with sizes less than 10 nm are dispersed on tungsten-based nanorods, and two sets of lattice fringes with spacing values of 0.22 and 0.34 nm can be seen

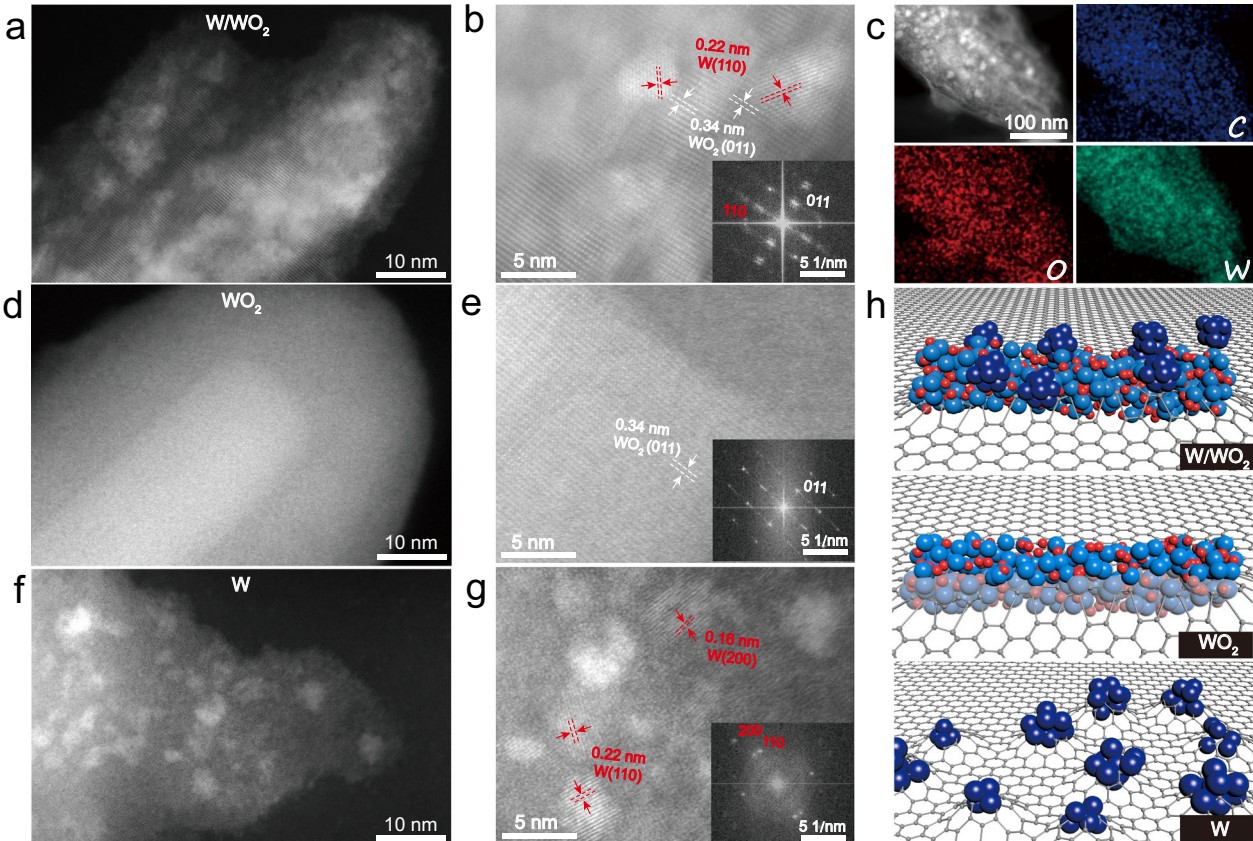

**Fig. 1 | Morphological characterizations of W/WO₂, WO₂ and W counterparts.** Low-magnification STEM images of **a** W/WO₂, **d** WO₂, and **f** W NPs. High-resolution STEM images of **b** W/WO₂, **e** WO₂, and **g** W NPs, with the corresponding FFT patterns in the insets. **c** STEM-EDS mapping of W/WO₂ showing the homogeneous distribution of C (blue), O (red) and W (green) elements. **h** Structural illustrations of W/WO₂ (blue and deep blue), WO₂ (blue) and W NPs (deep blue) embedded on carbon matrix.

from the high-magnification STEM image (Fig. 1b). Combining the analysis of corresponding fast Fourier transform (FFT) pattern (inset in Fig. 1b), we can conclude that the observed lattice fringes should be attributed to the (110) and (011) facet of supported W NPs and underlying WO₂ matrix, respectively. Meanwhile, the homogeneous distribution of C, O, and W elements over the entire W/WO₂ metallic heterostructure is visualized by the STEM-EDS mapping images (Fig. 1c). In addition, the morphology of WO₂ nanorods and W NPs were also examined. Figure 1d shows a typical carbon-encapsulated WO₂ nanorod, and the corresponding high-magnification STEM image and FFT pattern evidence the single-crystalline phase of WO₂ species (Fig. 1f). In contrast, numerous W NPs with lateral dimension below 5 nm are immobilized on carbon matrix (Supplementary Fig. 6, Fig. 1f, g), indicating the 1D $W_{18}O_{49}$ precursors have been disintegrated into ultrasmall W NPs under high-temperature reducing conditions. Therefore, based on above crystal structure and morphological characterizations, three types of tungsten-based catalysts (W, WO₂, and W/WO₂) are indeed obtained (Fig. 1h).

**Structural characterizations of W/WO₂ metallic heterostructure**

X-ray photoelectron spectroscopy (XPS) and X-ray adsorption spectroscopy (XAS) were performed to gain insight into the subtle change of the local chemical and electronic structures for the above three tungsten-based materials in. As shown in Fig. 2a, W and WO₂ samples exhibit $W^0$ and $W^{4+}$ signals at low (31.4 eV for $W^0$ $4f_{7/2}$ and 33.5 eV for $W^0$ $4f_{5/2}$) and high (32.7 eV for $W^{4+}$ $4f_{7/2}$ and 34.8 eV for $W^{4+}$ $4f_{5/2}$) binding energies after deconvolution, respectively. In addition to the intrinsically characteristic peaks, the inevitable surface high-valence oxidation species (35.5 eV for $W^{6+}$ $4f_{7/2}$ and 37.6 eV for $W^{6+}$ $4f_{5/2}$) are also observed

on W and WO₂ samples[31]. As expected, W/WO₂ sample contains features of both W and WO₂ species at the same binding energies after deconvolution, thus demonstrating the coexistence of W and WO₂ components in W/WO₂. The normalized X-ray absorption near-edge structure (XANES) profiles of W $L_3$-edge reveal that the white line intensity of W/WO₂ sample is much higher than that of metallic W NPs, but slightly lower relative to that of WO₂ counterpart (Fig. 2b), indicating the average oxidation state of tungsten atoms in W/WO₂ is between 0 and 4 + . Also, the whole W $L_3$-edge XANES profile of W/WO₂ extremely resembles that of WO₂ rather than metallic W, further suggesting that W atoms of WO₂ and W/WO₂ materials may have similar chemical state[32]. Correspondingly, from the viewpoint of coordinated oxygen atoms, the O K-edge near edge X-ray absorption fine structure (NEXAFS) spectroscopy can give additional information on the chemical structures of W atoms in above-mentioned tungsten species (Fig. 2c). One can see that both WO₂ and W/WO₂ samples exhibit split peaks located at energies ranging from 530 to 535 eV, where the peak at low energy (530.7 eV) originates from the overlapping band between W$5d$ and O$2p$ orbitals, corresponding to the W-O bonds, while the superoxide ($O_2^-$) species neighboring to surface oxygen vacancies should be responsible for another one at high energy (532.2 eV)[33]. Comparing to the O K-edge NEXAFS spectroscopy of WO₂ counterpart, W/WO₂ metallic heterostructure exhibits decreased intensity of W-O bonds but increased signal of $O_2^-$ species, indicating the formation of rich oxygen vacancies. Meanwhile, such an enrichment of oxygen vacancies in W/WO₂ metallic heterostructure is also resolved by electron spin resonance (ESR) spectroscopy (Supplementary Fig. 7), which exhibits a strong and symmetrical ESR signal at g = 2.002, manifesting rich unpaired electrons trapped on vacancies around W centers[34].

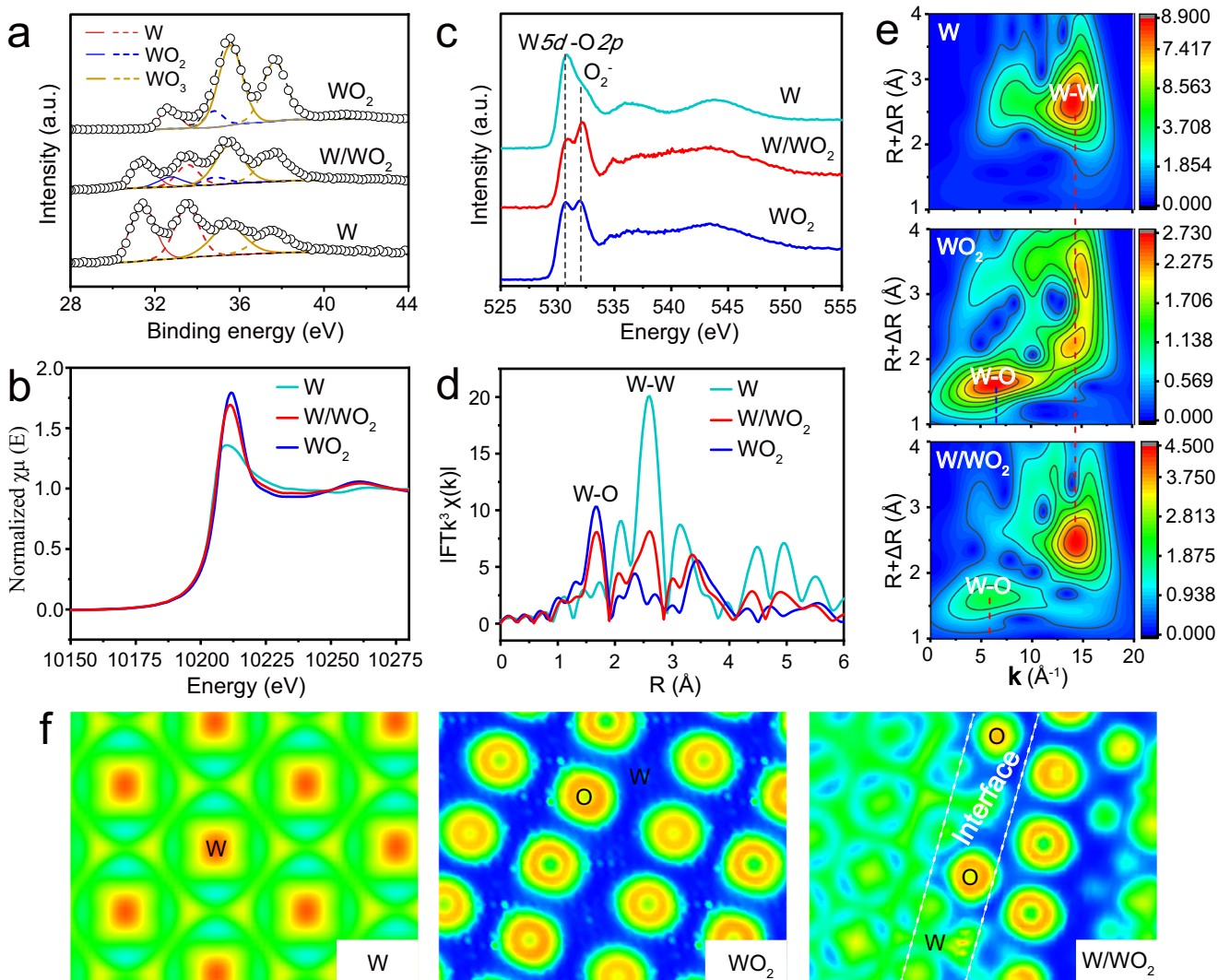

**Fig. 2 | Structural characterizations of W/WO₂, W and WO₂ counterparts. a** W $4f$ core-level XPS spectra. **b** W $L_3$-edge XANES spectra. **c** O $K$-edge NEXAFS spectra. **d** R space profiles of W (light blue), WO₂ (blue), and W/WO₂ (red) extracted from the corresponding W $L_3$-edge EXAFS spectra. **e** WT-EXAFS plots. **f** The calculated ELF images of W, WO₂, and W/WO₂. Green to red indicates the gradually increased electron localization.

Further, the subtle difference of local coordination structure in W, WO₂, and W/WO₂ samples were discriminated by the Fourier-transformed extended X-ray absorption fine structure (FT-EXAFS) spectra (R-space) (Fig. 2d). As can be seen, W sample exhibits a distinct peak at 2.6 Å, corresponding to the W-W scattering path, while a predominant peak at approximately 1.6 Å is presented on WO₂ sample, which can be assigned to the W-O coordination. Meanwhile, we also observe a weak but still visible W-W coordination at 2.4 Å on WO₂ sample, which is almost absent in stoichiometric WO₃ reference (Supplementary Fig. 8), demonstrating the metallic property of WO₂ phase. As expected, the R-space profile of W/WO₂ heterostructure exhibits the combined features of W and WO₂ counterparts. However, comparing to WO₂ sample, W/WO₂ sample exhibits a relatively weak W-O coordination but significantly increased intensity of W-W scattering path, implying the introduction of W NPs probably breaks the homogeneity of local W-O coordination structure, and further enhance the metallic property of WO₂ matrix. Wavelet transform (WT)-EXAFS with high-resolution in both **k** and R spaces can be used to visually examine the atomic configuration of W atoms. In line with the FT-EXAFS analysis, the WT-EXAFS contour plot of W/WO₂ sample shows an increased intensity of W-W scattering path centered at about **k** = 14.3 Å⁻¹ relative to that of WO₂ sample, further suggesting the

increased metallic feature. Moreover, W/WO₂ sample exhibits intensity maximum at **k** = 5.6 Å⁻¹ for W-O coordination, which is definitely different from WO₂ sample (**k** = 6.3 Å⁻¹), suggesting the local W-O bonding configuration of WO₂ matrix has been altered due to the introduction of W NPs. Finally, the electronic structures of three tungsten-based materials were also examined by the electron localization function (ELF) calculations. Unlike the uniform electronic structures of W and WO₂ counterparts, W/WO₂ model exhibits distinctly modified electronic structures at the interface, where the coexistence of electron localized (O atoms in WO₂) and delocalized (W⁰ atoms in W metal) states can afford W/WO₂ heterostructure with synergistic effort for kinetically fast water dissociation and hydrogen desorption steps in alkaline HER process.

## Evaluation of alkaline HER electrocatalysis using W/WO₂ catalyst

The HER performance of free-standing W/WO₂ solid-acid catalyst was evaluated in 1.0 M KOH electrolyte using a typical three-electrode setup, where bare carbon coated Ni foam (C@Ni), W NPs, WO₂, and PtRu/C catalysts were also measured under the same conditions for comparison. As shown in Fig. 3a, bare C@Ni sample exhibits a negligible alkaline HER activity, although W and WO₂ catalysts exhibit

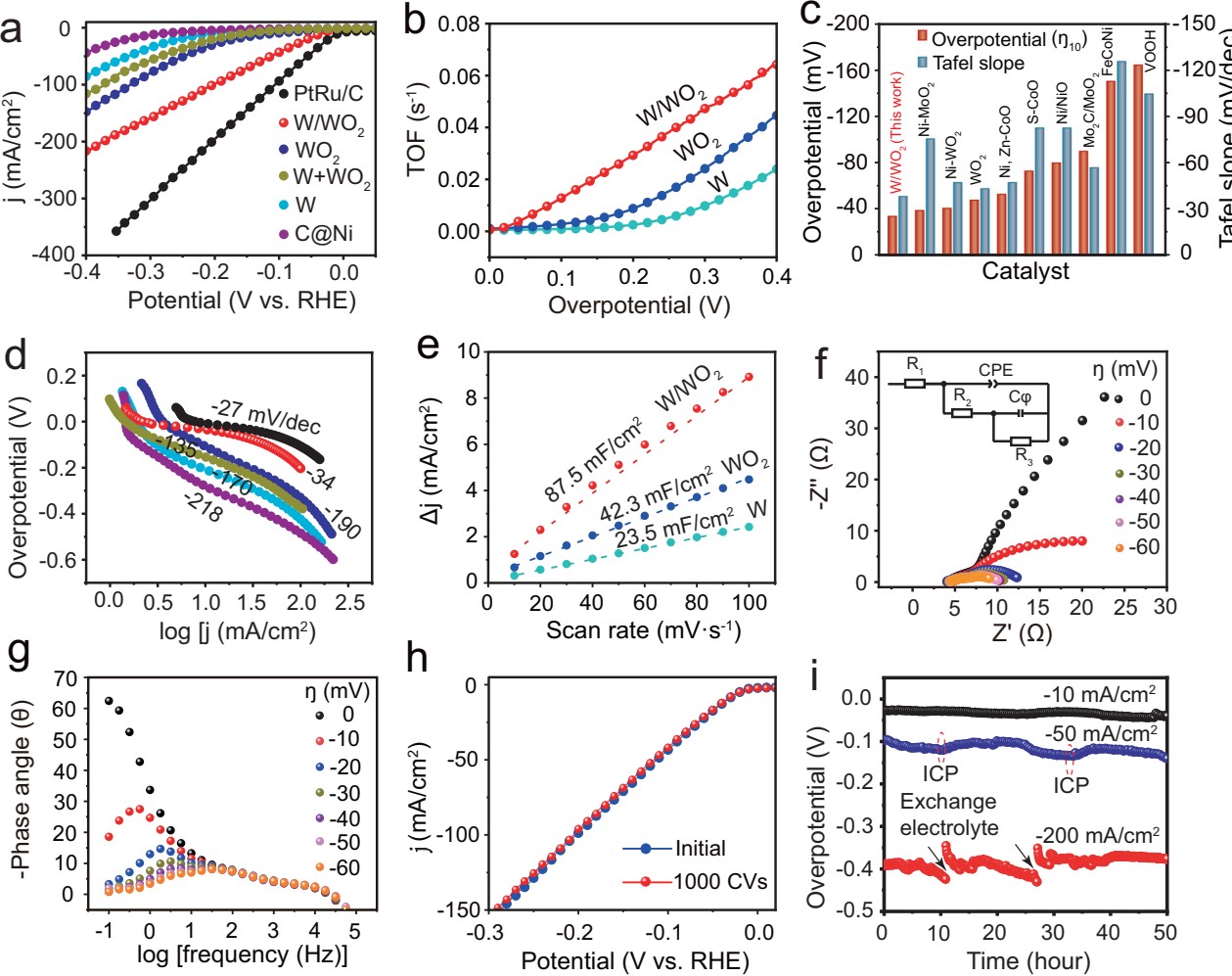

**Fig. 3 | The evaluation of HER performance of $W/WO_2$ solid-acid catalyst in 1.0 M KOH electrolyte (pH = 14, Rs = ~4.0 Ω, mass loading = 3.2 mg/cm²).** **a** polarization (LSV) curves of C@Ni (wine red), W (light blue), $WO_2$ (deep blue), $W + WO_2$ (green), $W/WO_2$ (red) and commercial PtRu@C (black) catalysts. **b** TOF plots of W, $WO_2$, and $W/WO_2$ catalysts at overpotentials of 0 - 0.4 V. **c** Comparison of overpotentials (10 mA/cm²) and Tafel slopes of $W/WO_2$ solid-acid catalyst and previously reported excellent transition-metal-oxide based HER catalysts in alkaline electrolyte (see Supplementary Table 1). **d** Tafel plots of C@Ni (wine red), W (light blue), $WO_2$ (deep blue), $W + WO_2$ (green), $W/WO_2$ (red) and commercial PtRu@C (black) catalysts. **e** The

determination of $C_{dl}$ by plotting the current density variation (Δj) against the scan rates (10–100 mV s⁻¹). **f** Nyquist plots (the inset shows the equivalent circuit for the simulation) and **g** the corresponding Bode phase plots of $W/WO_2$ solid-acid catalyst with the increase of applied overpotentials at 0 (black), −10 (red), −20 (deep blue), −30 (green), −40 (purple), −50 (light purple), and −60 (orange) mV. **h** LSV curves of $W/WO_2$ solid-acid catalyst before (deep blue) and after (red) 1000 CVs. **i** Chronopotentiometry measurements of $W/WO_2$ solid-acid catalyst at current densities of −10 (black), −50 (blue), and −200 (red) mA/cm², and the red dashed circle indicates the extraction of electrolyte for ICP detection.

enhanced alkaline HER activities with decreased overpotentials at −10 mA/cm² (W: $\eta_{10}$ = −183 mV; $WO_2$: $\eta_{10}$ = −106 mV), their activities are still inferior to other W-/Mo-based catalysts and conventional 3d-metal (Fe, Co, Ni) oxides reported to date (Fig. 3c and Supplementary Table 1). Encouragingly, the as-obtained $W/WO_2$ solid-acid catalyst displays a superior alkaline HER activity with value of $\eta_{10}$ as low as −35 mV, excelling all previously reported tungsten oxide catalysts and most state-of-the-art non-noble oxide catalysts (Fig. 3c and Supplementary Table 1), and even being comparable to commercial PtRu/C (−11 mV). In addition, the alkaline HER activity of the physical mixture of W and $WO_2$ (W + $WO_2$) was also examined under the same conditions, where the values of $\eta_{10}$ and Tafel slope are determined to be −153 mV and −135 mV/dec, respectively, and the markedly improved Tafel slope may be attributed to the synergistic effect of $WO_2$ and W components for water dissociation and hydrogen desorption steps in alkaline HER process. However, the alkaline HER activity of W + $WO_2$ is still inferior to $W/WO_2$ catalyst, suggesting the $W/WO_2$ interfaces constructed by chemical bonds can improve the alkaline HER activity intrinsically. The turnover frequency (TOF) plot of $W/WO_2$ solid-acid

catalyst has been calculated at overpotentials of 0 - 0.4 V (Fig. 3b, calculated detail seen in method section), where $W/WO_2$ solid-acid catalyst exhibits a high TOF value of 0.013 s⁻¹ at overpotential of −100 mV, nearly 4.6- and 15.1-fold higher than $WO_2$ and W counterparts, demonstrating the high intrinsic activity of $W/WO_2$ solid-acid catalyst. Further, the accelerated reaction kinetics of alkaline HER process on $W/WO_2$ solid-acid catalyst was revealed by Tafel slope. $W/WO_2$ solid-acid catalyst exhibits a sharply decreased Tafel slope (−34 mV/dec) relative to those of W (−170 mV/dec) and $WO_2$ (−190 mV/dec) counterparts (Fig. 3d), even being comparable to the value of commercial PtRu/C catalyst (−27 mV/dec). This result indicates the high energy barrier of additional water dissociation has been substantially weakened, and the concomitant hydrogen production follows the fastest kinetics of Tafel pathway due to the coverage of rich protons on $W/WO_2$ catalyst surface. Moreover, the HER performance of W, $WO_2$, and $W/WO_2$ powders were also examined using the rotating disk electrode technique in 1 M KOH electrolyte (Supplementary Fig. 9). As can be seen, $W/WO_2$ catalyst still exhibits a remarkable alkaline HER activity with a low overpotential (−60 mV) at −10 mA/cm² and a small Tafel

slope (−54 mV/dec), which are more excellent than those of W and $WO_2$ counterparts, and still excelling most previously reported metal oxides (Fig. 3c).

In order to better understand the proton coverage on $W/WO_2$ catalyst surface, double-layer capacitance ($C_{dl}$) and electrochemical impedance spectroscopy (EIS) of $W/WO_2$ catalyst were examined, with W and $WO_2$ counterparts as references. The larger value of $C_{dl}$ usually implies the higher electrochemical surface area (ECSA) for alkaline HER process. The $C_{dl}$ value of catalyst can be extracted from the cyclic voltammetry (CV) curves with different scan rates at non-Faradaic voltage windows (Supplementary Fig. 10). The $C_{dl}$ value of $W/WO_2$ catalyst is determined to be up to 87.5 mF/cm$^2$, nearly 3.7- and 2.1-fold enhancement than those of W and $WO_2$ counterparts, respectively (Fig. 3e), suggesting the construction of $W/WO_2$ metallic heterostructure can provide rich active sites for the adsorption of $H_2O$ reactants and reaction intermediates in alkaline HER process[35]. The electrochemical impedance spectroscopy (EIS) measurements were then performed to track the accumulation of protons deriving from the activation of $H_2O$ reactants on $W/WO_2$ catalyst surface[36]. All Nyquist plots of $W/WO_2$ samples were simulated by a double-parallel equivalent circuit model in accordance with previous reports[36,37], where $R_1$ represents the uncompensated solution resistance ($R_s$), the first parallel components (constant phase element (CPE) and $R_2$) indicate the charge transfer resistance caused by the adsorption and activation of water molecules at low frequencies[38,39], and the second parallel ones of $R_2$ and $C_\varphi$ are attributed to the hydrogen adsorption resistance and pseudo-capacitance at high frequencies, respectively (Fig. 3f and Supplementary Table 2). As expected, all electrochemically treated $W/WO_2$ samples exhibit the similar $R_1$ value (~4.0 Ω), and the small values of $R_2$ for all $W/WO_2$ catalysts suggest the fast charge transfer kinetics between catalyst surface and $H_2O$ molecules. In particular, $R_2$ decreases to 4.4 Ω sharply with a negligible water diffusion resistance at applied overpotential of −20 mV, indicating the adsorption and activation of water molecules on the $W/WO_2$ catalyst surface can be achieved under low overpotentials. Further, we also notice that $R_3$ and $C_\varphi$ are largely overpotential-dependent, where $W/WO_2$ catalysts exhibit significantly decreased $R_3$ with increased $C_\varphi$ when increasing the applied overpotentials, in particular, the value of $R_3$ can be as low as approximately 1.8 Ω, while $C_\varphi$ is up to 0.017 F at an overpotential of −30 mV, suggesting the hydrogen adsorption resistance is very small, and the pseudo-capacitance of proton coverage is very large on $W/WO_2$ catalyst surface under low overpotentials. In contrast, both W and $WO_2$ counterparts show a sluggish alkaline HER kinetics with large values of $R_{ct}$ and high negative phase angles on the Nyquist and Bode plots, respectively (Supplementary Fig. 11), suggesting a sluggish HER kinetics in an alkaline environment.

Besides of high electrocatalytic activity, catalyst durability is another significant concern for practical application. In order to evaluate the durability of the $W/WO_2$ solid-acid catalyst, the as-obtained catalyst was firstly performed by 1000 cyclic voltammograms (CVs) in the voltage window from 0 to −0.2 V (versus RHE) with a scan rate of 100 mV/s. As can be seen, the HER polarization curve of $W/WO_2$ almost overlaps with the original one after 1000 potential cycles (Fig. 3h), suggesting the stable proton-coupled electron redox activity of $W/WO_2$ solid-acid catalyst in alkaline electrolyte. A prolonged chronopotentiometry measurement was further applied to evaluate the long-term durability of the catalysts at current densities of −10, −50 mA/cm$^2$. It can be seen that the superior alkaline HER activity is well retained on $W/WO_2$ solid-acid catalyst after continuous hydrogen production for more than 50 h. The relatively good stability of $W/WO_2$ solid-acid catalyst in alkaline electrolyte is also confirmed by the inductively coupled plasma-optical emission spectroscopy (ICP-OES). After long-term hydrogen production, the initial electrolyte, used electrolyte, and refreshed electrolyte (indicated by red dashed circle) shows no significant

increase in the concentrations of dissolved W species with values of $7.5 \times 10^{-6}$, $9.8 \times 10^{-6}$, and $6.3 \times 10^{-7}$ mol/L, respectively. In particular, the rather low concentration ($6.3 \times 10^{-7}$ mol/L) of dissolved W species in the refreshed electrolyte directly suggests the structural robustness of $W/WO_2$ catalyst after long-term hydrogen production in alkaline electrolyte. In addition, we also evaluate the stability of $W/WO_2$ catalyzing alkaline HER process at a current density of −200 mA/cm$^2$ (the lower limit for industrial water electrolysis). No significant activity loss can be observed on $W/WO_2$ catalysts after long-term hydrogen production, suggesting the good catalytic stability at industrial current density. For the characterization of used catalyst, a rough catalyst surface with rich defects is observed on the used $W/WO_2$ solid-acid catalyst (Supplementary Fig. 12), suggesting the insertion of hydrogen may alter the local chemical structure of $W/WO_2$ solid-acid catalyst. Meanwhile, the enhanced signal of oxygen vacancies detected by ESR and O K-edge NEXAFS demonstrates that the proton-coupled electron reaction of HER process might have caused a slight reduction of underlying $WO_2$ matrix (Supplementary Fig. 13). Therefore, based on the comprehensive evaluations of alkaline HER activity and stability on $W/WO_2$ materials, the significantly improved structural robustness of our well designed $W/WO_2$ composites in high-pH solutions can be attributed to the following two reasons: (i) $W/WO_2$ heterostructures have intrinsically good oxidation resistance with co-existence of metal and oxide features and the protection of surface carbon layers[19,29,40,41], meanwhile, the strong chemical and electronic interactions of W and $WO_2$ components within $W/WO_2$ may further improve the structural robustness in alkaline solutions[42,43]; (ii) unlike the naturally alkaline leaching under circuit potential condition, the negative potentials at cathode can provide rich electrons to avoid the oxidation and dissolution of low-valence tungsten species during alkaline HER process.

## Multiple spectroscopy characterizations discovering electrocatalytic mechanism

Comprehensive characterizations were performed to gain insight into the origin of significantly enhanced alkaline HER activity on $W/WO_2$ metallic heterostructure. Firstly, the deuterated-effect-induced inferior alkaline HER activity suggests that the fast HER kinetics of water dissociation has been lowered on $W/WO_2$ catalyst surface because of the larger zero-point energy of OD* relative to the OH* intermediates (difference of ~1400 cal) (Supplementary Fig. 14a)[44]. Besides, we also observe that the deuterated effect has more negative influence on the Tafel slope of $W/WO_2$ catalyst relative to W and $WO_2$ counterparts (Supplementary Fig. 14b), because the sluggish kinetics of water dissociation hampers the proton coverage on $W/WO_2$ catalyst surface, which severely lowers subsequent hydrogen desorption kinetics. Secondly, near ambient pressure X-ray photoelectron spectroscopy (NAP-XPS) measurement was performed to investigate the adsorption and cleavage of $H_2O$ molecules on W, $WO_2$, and $W/WO_2$ catalyst surface under a water pressure of 0.1 mbar (Fig. 4a), with the core-level XPS spectroscopy recorded under an ultrahigh vacuum (UHV) condition as a reference[8]. For the referenced O $1s$ XPS profile of $W/WO_2$ sample under an UHV condition. It exhibits two peaks at 530.3 and 531.3 eV after deconvolution (Fig. 4b), corresponding to the lattice oxygen (W-O) species and oxygen vacancies ($O_v$), respectively[45]. After introducing 0.1 mbar $H_2O$ molecules (Supplementary Fig. 15), $W/WO_2$ metallic heterostructure exhibits a vanished $O_v$ signal but significantly enhanced concentrations of W-OH (532.3 eV) and $H_2O$ (533.3 eV) species at higher binding energies (Supplementary Table 3), indicating $O_v$ have served as the adsorption sites of $H_2O$ molecules, and the cleavage of H-OH bonds is easily proceeded on $W/WO_2$ catalyst surface. Also, such an accelerated water dissociation kinetics of $W/WO_2$ catalyst has been confirmed by the obvious decrease of low-valence tungsten species in the corresponding W $4f$ core-level XPS spectra (Fig. 4c). In

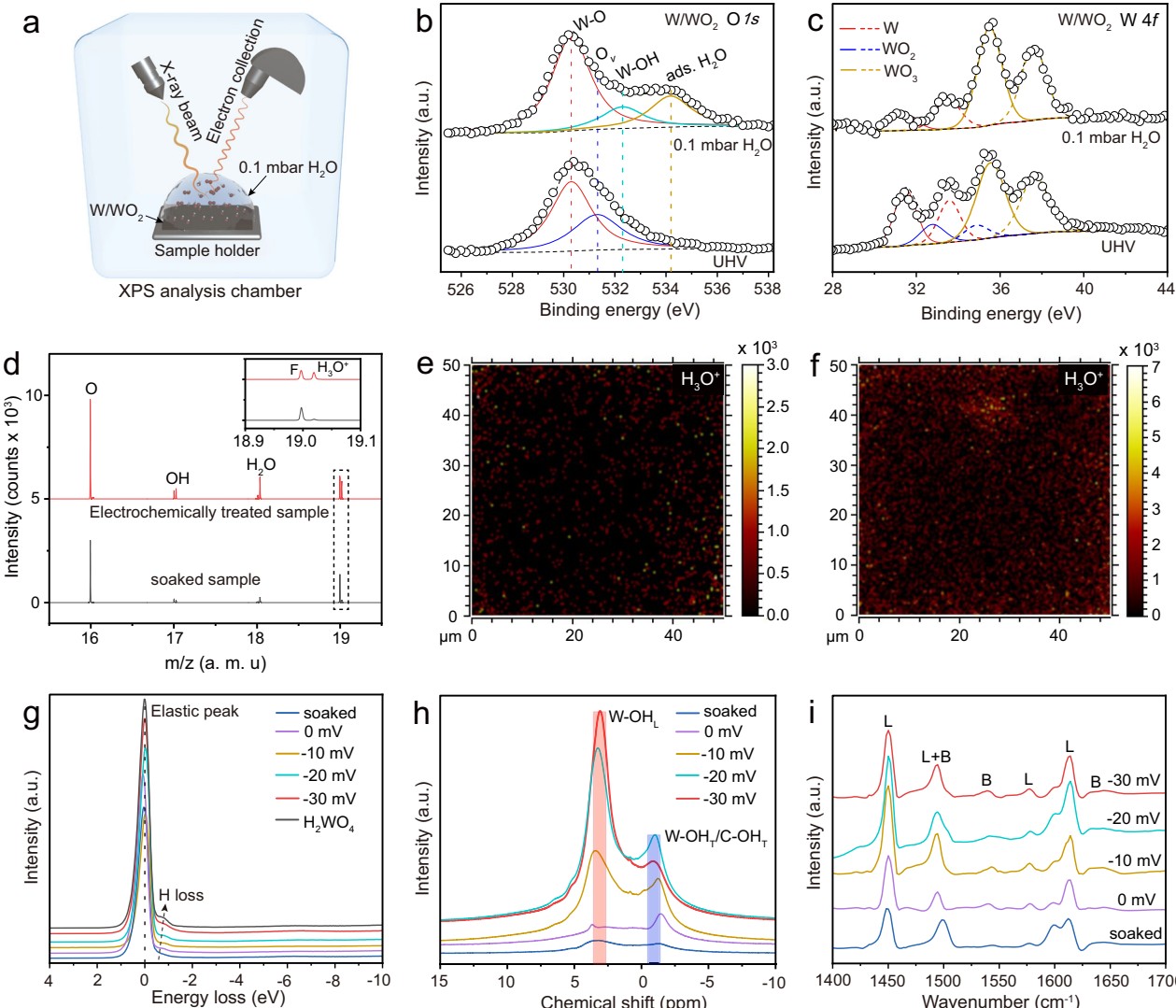

**Fig. 4 | Multiple high-resolution spectroscopy characterizations tracing the coverage and chemical state of protons on W/WO₂ solid-acid catalyst.** **a** Schematic illustration of the in situ NAP-XPS measurement under 0.1 mbar H₂O atmosphere. **b** O 1s and **c** W 4f XPS spectra of W/WO₂ recorded under UHV and 0.1 mbar H₂O conditions, respectively. **d** TOF-SIMS profiles of W/WO₂ reference (black) and electrochemically treated sample (red), and the inset shows the enlarged signals ranging from 18.9 to 19.1, where the fluorine (F, m/z = 19.00) signal may be originated from the inevitable contamination of fluorine-containing sealing ring in

the analysis chamber during Bi³⁺ cations impacting the sample, while the H₃O⁺ species can be responsible for the higher signal at m/z = 19.02. **e** 2D distribution images of H₃O⁺ species on W/WO₂ only soaked in alkaline electrolyte and **f** W/WO₂ after alkaline HER process. **g** REELS spectra, **h** ¹H MAS NMR, and **i** Py-IR spectra of W/WO₂ catalysts after electrochemical treatments by the gradually increased overpotentials, where soaked, 0, -10, -20, and -30 mV are represented by deep blue, pink, orange, light blue, and red colors, and the black color stands for commercial H₂WO₄ sample.

contrast, W and WO₂ counterparts display inferior activities of water dissociation kinetics (Supplementary Fig. 16 and Supplementary Table 3), in particular, only an extremely weak W-OH signal is observed on W catalyst surface after exposure in 0.1 mbar H₂O atmosphere, because W⁰ (W) sites with more filled d orbitals have weaker electrostatic affinity to electron-rich H₂O molecules than W⁴⁺ (WO₂) atoms[46].

Finally, with the clear elucidation of proton production on W/WO₂ catalyst surface in water dissociation step, the coverage and chemical environment of produced protons determine the activity of subsequent hydrogen generation on W/WO₂ solid-acid catalyst surface[36]. Time-of-flight secondary ion mass spectroscopy (TOF-SIMS) is very sensitive to the examine the hydrogen species after alkaline HER process. In order to eliminate the interference of adsorbed H₂O molecules and OH⁻ anions (electrolyte), W/WO₂ sample only soaked in KOH electrolyte was selected as a reference. We can observe that the electrochemically treated W/WO₂ catalyst exhibits markedly increased signals of hydrogen-containing intermediates, i. e., OH, H₂O, H₃O⁺, in

particular, the sharply increased H₃O⁺ signal directly suggests the formation of proton-concentrated catalyst surface[21] (Fig. 4d). In order to visually display the distribution of H₃O⁺ on the electrochemically treated W/WO₂ catalyst surface, two-dimensional (2D) image analysis of the soaked and electrochemically treated W/WO₂ samples are also provided in Fig. 4e, f. Obviously, one could notice that the concentration of H₃O⁺ species on the electrochemically treated W/WO₂ catalyst is much higher than that of the only soaked counterpart, indicating the achievement of proton-concentrated catalyst surface on W/WO₂ solid-acid catalyst. Multiple spectroscopy characterizations were further performed to analyze the property of the produced protons on the used W/WO₂ catalyst surface treated by increased applied overpotentials (η=0, −10, −20, −30 mV), with W/WO₂ sample only soaked in KOH electrolyte and commercial tungstic acid (H₂WO₄) as the references. The reflection electron energy loss spectroscopy (REELS) can break the limitation of conventional XPS technique for the detection of H element. As shown in Fig. 4g, the REELS plot of W/WO₂

soaked sample only exhibits a predominantly elastic peak, while an obvious H signal neighboring to elastic peak is observed on commercial $H_2WO_4$ sample[47–49], indicating the hydrogen concentration of W/$WO_2$ soaked sample is lower than the detecting limitation (molar ratio: ~20%) of REELS technique. As expected, all electrochemically treated W/$WO_2$ catalysts exhibited gradually enhanced H signals when increasing the applied overpotentials (0 to −30 mV), indicating a proton-concentrated surface could be constructed on W/$WO_2$ metallic heterostructure under an ultra-low overpotential, which is in good consistency with the analysis of EIS measurement. Note that the major contributor of detected H signal should be attributed to W/$WO_2$ species rather than underlying carbon matrix, as evidenced by the REELS measurements of bare carbon materials after treatments under increased overpotentials (Supplementary Fig. 17). Solid-state $^1H$ magic-angle-spinning nuclear magnetic resonance ($^1H$ MAS NMR) spectra exhibits two broad bands at approximately −1.3 and 3.4 ppm on soaked W/$WO_2$ sample (Fig. 4h), in which the peak at lower chemical shift can be attributed to the mixture of terminated hydroxyl (W-$OH_T$ and/or C-$OH_T$) species on $WO_2$ and carbon matrix, while the higher one originates from the lattice hydroxyl (W-$OH_L$) species in $WO_2$ matrix. Similar to REELS characterization, all electrochemically treated W/$WO_2$ samples exhibit significant increased signals of W-$OH_T$ and W-$OH_L$ species under increasing overpotentials, indicating the increased coverage of protons over W/$WO_2$ catalyst surface. Meanwhile, we also notice that the proportion of W-$OH_T$/W-$OH_L$ decreases with the increase of applied potentials, which may be attributed to the continuously increased coverage of produced protons. The increasing negative potentials can result in more water molecules being adsorbed on W/$WO_2$ catalyst surface, and the water molecules rapidly react with electrons for the cleavage of H-OH bonds, then the produced H* intermediates insert into the lattice of tungsten oxides (W-$OH_L$) for constructing proton-concentrated catalyst surface with electronic and chemical environments approaching to those of $H_2WO_4$ materials. While the terminated electron-rich hydroxyl species, such as the produced OH* intermediates and chemically/physically adsorbed $OH^-$ molecules (W-$OH_T$) cannot follow the increasing tendency because of the enhanced Coulomb effect at cathode. In addition, $^1H$ MAS NMR profiles collected at overpotentials of −10, −20, and −30 mV become extremely similar to that of commercial $H_2WO_4$ reference (Supplementary Fig. 18), demonstrating the chemical environment of protons

on W/$WO_2$ catalyst surface is approaching to commercial $H_2WO_4$ reference. All Py-IR spectroscopy of electrochemically treated samples exhibit typical adsorption peaks in the wavenumber range of 1400 ~ 1650 $cm^{-1}$, where peaks at 1450, 1577, and 1613 $cm^{-1}$ are ascribed to the chemisorption of pyridine at surface Lewis acid sites (L), while peaks at 1540 and 1638 $cm^{-1}$ indicate the presence of Brønsted acid sites (B), and the signal at 1494 $cm^{-1}$ contains contribution of Lewis acid and Brønsted acid sites (Fig. 4i and Supplementary Fig. 19). Impressively, the electrochemically treated W/$WO_2$ catalysts also exhibited enhanced concentrations of Brønsted acid sites with the increase of applied overpotentials (Supplementary Table 4), indicating the produced protons of water dissociation mainly serve as Brønsted acids with reversible behaviors of hydrogen adsorption and desorption on W/$WO_2$ solid-acid catalyst surface[26,27].

## DFT calculations

Such a significant enhancement of alkaline HER activity of W/$WO_2$ solid-acid catalyst was also elucidated by first-principle density functional theory (DFT) calculations, with simulations of W and $WO_2$ models as references (Fig. 5a, Supplementary Fig. 20). Specifically, more detailed front, side and top structures of W/$WO_2$ interface were also calculated in Supplementary Fig. 21. $WO_2$ (−0.33 eV) and W/$WO_2$ (−0.32 eV) materials exhibit more negative adsorption energies of $H_2O$ molecules than that of W sample (−0.14 eV) (Fig. 5b), which suggests the $WO_2$ and W/$WO_2$ catalyst surfaces are beneficial for the adsorption and activation of $H_2O$ reactants[50,51]. As expected, W exhibits a sluggish kinetics of prior water dissociation step with energy barrier ($\Delta G_{H_2O}$) up to 0.84 eV, while the activation energy barrier of $H_2O$ molecules can be sharply reduced to 0.06 eV on $WO_2$ model (Fig. 5b), indicating oxygen-vacancy-rich $WO_2$ is generally effective for the cleavage of H-OH bonds. Encouragingly, the energy barrier of water dissociation is further decreased to 0.02 eV on W/$WO_2$ catalyst surface (Supplementary Table 5), excelling most previously reported noble metal alloy catalysts[52,53]. With an ultra-low energy barrier of water dissociation benefiting the formation of available protons on W/$WO_2$ catalyst surface, the following proton detachment will determine the reaction rate of alkaline HER process, where hydrogen adsorption free energy ($\Delta G_H$) of H* intermediates is a good descriptor to evaluate the HER activity, and the optimum $\Delta G_H$ is usually around thermoneutral value (0)[22]. In contrast to the calculations of water dissociation, W catalyst exhibits a

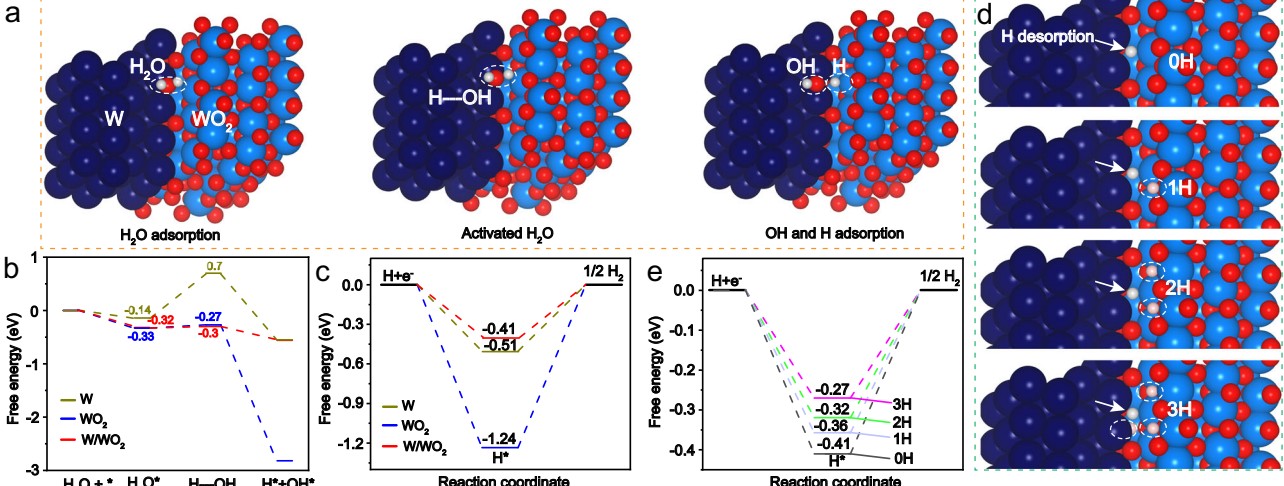

**Fig. 5 | DFT calculations of alkaline HER activities on W, $WO_2$, and W/$WO_2$ models. a** Schematic pathway for alkaline HER process on W/$WO_2$ interface, where $H_2O$ molecule undergoes water adsorption, activated $H_2O$ adsorption, produced OH and H adsorption in alkaline HER process. The corresponding calculated free energy diagrams for **b** water adsorption and dissociation, and **c** hydrogen

desorption steps on W/$WO_2$ interface, W (110) and $WO_2$ (01-1) facets, respectively. **d** H desorption (white) models and **e** the corresponding calculated free energies tuned by continuously increased neighboring Brønsted acid sites (pink). Panel **a** and **d** are created by VESTA software[54].

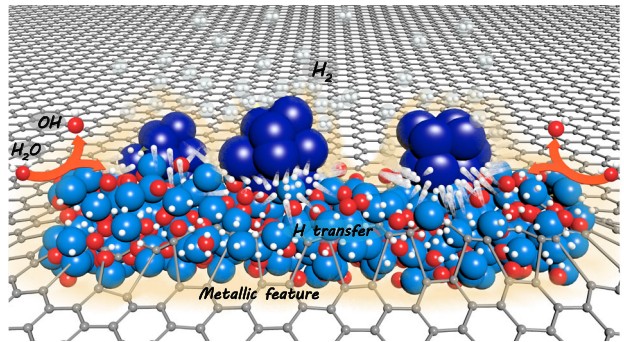

**Fig. 6 | Schematic illustration of the dynamic proton-concentrated catalyst surface.** The cleavage of water reactants is proceeded on $WO_2$ surface for the generation of protons (light blue surface), where the aggressive $OH^-$ species can be rapidly repulsed from the electron-rich catalyst surface (orange layer), while the highly active protons prefer to transfer to the interface of $WO_2$ (light blue) and W (deep blue) components for the proton recombination and release of hydrogen gas.

lower hydrogen adsorption energy (−0.51 eV) than that of $WO_2$ catalyst (−1.24 eV) (Fig. 5c), implying zero-valence W sites are more beneficial for hydrogen desorption. Accordingly, the calculated $\Delta G_H$ (−0.41 eV) decreases sharply when coupling W and $WO_2$ components as heterostructures (Fig. 5c), indicative of significantly improved hydrogen desorption kinetics on $W/WO_2$ catalyst surface. Encouragingly, we observe that the $\Delta G_H$ value (−0.27 eV) of $W/WO_2$ is extremely closer to thermoneutral value (0) after the $W/WO_2$ interface inserted by $H^+$ atoms ($W/H_xWO_y$) (Fig. 5d, e), indicating the neighboring Brønsted acid site can further improve the hydrogen desorption kinetics on $W/WO_2$ catalyst surface. Moreover, the hydrogen desorption activities of $W/WO_2$ interface can be further tuned by the amounts of neighboring Brønsted acid site (Fig. 5e), demonstrating the coverage of Brønsted acids also plays a vital role in the hydrogen desorption step, which agrees well with aforementioned experimental characterizations (Figs. 3a, e, f and 4d, e, f).

Based on above comprehensive characterizations in solid experimental evidences and theoretical simulations, a well-defined dynamic proton-concentrated surface is constructed on $W/WO_2$ solid-acid catalyst, where the dynamic feature of proton-concentrated surface facilitates the proton transfer for the cycling of active sites on catalyst surface (Fig. 6). First, the oxygen-vacancy-rich $WO_2$ component mainly serves as highly active Lewis acid sites for the adsorption and cleavage of $H_2O$ molecules (proton generation). Second, protons can be enriched on $W/WO_2$ catalyst surface under ultra-low overpotentials due to the strong hydrogen storage ability of $H_xWO_y$ intermediates (Brønsted acid sites, proton storage). Finally, the appealing electronic interaction between zero-valence W atoms and protons accelerates the deprotonation kinetics of Brønsted acids for the cycling of active sites (proton donation and regeneration of active sites).

## Discussion
In summary, a facial pyrolysis-reduction method is used to prepare free-standing $W/WO_2$ solid-acid catalyst on Ni foam. Owing to the in situ constructed solid-acid catalyst surface with reversible adsorption/desorption behaviors of protons in alkaline HER process. The well-designed $W/WO_2$ heterostructure catalyzing hydrogen production follows the kinetically fast Volmer-Tafel pathway with an ultra-low overpetential at −10 mA/cm² and a small Tafel slope (−34 mV/dec), as well as a long-term electrocatalytic stability (>50 h) in alkaline electrolyte, outperforming all tungsten/molybdenum oxide catalysts and most of 3d-metal oxides reported to date. Moreover, multiple spectroscopy characterizations combined with DFT calculations discover

that the dynamic proton-concentrated surface can be constructed on $W/WO_2$ metallic heterostructure under low overpotentials, enabling $W/WO_2$ catalyzing alkaline hydrogen production to follow kinetically fast Volmer-Tafel pathway. Our strategy of solid-acid catalyst and utilization of multiple spectroscopy characterizations may provide a valuable route for designing advanced all-non-noble catalytic system towards boosting HER performance in alkaline electrolyte.

## Methods
### Chemicals
Tungsten chloride ($WCl_6$ 99%, Aladdin, USA), Polyvinylpyrrolidone (PVP, Macklin, China), polyethylene oxide-co-polypropylene oxide-co-polyethylene oxide ($P_{123}$ 98%, Adamas Reagent, China), dopamine hydrochloride (DA 98%, Aladdin, USA), and 2-amino-2-hydroxymethyl-propane-1,3-diol (Tris 98%, Aladdin, USA).

### Preparation of $W_{18}O_{49}$ NWs
In a typical procedure, a mixture of $WCl_6$ (500 mg) and PVP (20 mg) precursors was dissolved in absolute ethanol (50 mL), and a homogeneous yellow solution was obtained after stirring for 30 min. Afterwards, the obtained solutions were transferred into a 100 mL Teflon-lined autoclave and heated at 180 °C for 24 h. The resultant blue $W_{18}O_{49}$ product was collected after purification and centrifugation.

### Preparation of $W_{18}O_{49}$@Ni foam
The Ni foam (2 cm × 4 cm) was treated by 0.1 M HCl solutions, and washed with absolute ethanol and distilled water before drying at 60 °C. A total of 50 mg $W_{18}O_{49}$ powder was re-dispersed into 2 mL ethanol by sonication, and the homogeneous ink was dropped onto the pre-treated Ni foam (2 cm × 4 cm). In order to better lock the $W_{18}O_{49}$ precursor, Ni foam supported $W_{18}O_{49}$ powder was pressed into a thinner foil structure (Supplementary Fig. 3).

### Preparation of $W_{18}O_{49}$@PDA@Ni foam
The fabricated $W_{18}O_{49}$@Ni foam was immersed into mixture solutions (250 mL) of DA (100 mg), $P_{123}$ (160 mg), and Tris (40 mg), and $W_{18}O_{49}$@PDA@Ni foam was obtained after stirring for 24 h. In addition, $W_{18}O_{49}$ powder was also added into the same mixture solutions containing DA, $P_{123}$, and Tris components with continuous stirring for 24 h. After that, $W_{18}O_{49}$@PDA powder was collected by purification and centrifugation.

### Preparation of $W/WO_2$ metallic heterostructure
Both $W/WO_2$ powder and free-standing electrode were prepared in accordance with the following thermal treatment. The desired $W/WO_2$ metallic heterostructure was obtained by pyrolyzing the $W_{18}O_{49}$@PDA@Ni foam ($W_{18}O_{49}$@PDA powder) precursor at 700 °C under a mixture of Ar/$H_2$ (v/v = 8:1) atmosphere for 2 h. For comparison, power and free-standing types of $WO_2$ sample were synthesized at 700 °C under a pure Ar atmosphere for 2 h, while W sample was collected at high temperature of 750 °C under a mixture of Ar/$H_2$ (v/v = 8:1) atmosphere for 2 h. The total loading mass of supported materials ($W/WO_2$ and C, 32 mg) can be determined by the mass difference of bare Ni foam (2 × 4 cm, 250 mg) and Ni foam supported $W/WO_2$ materials (2 × 4 cm, 282 mg). The loading mass of tungsten element was 3.2 mg/cm² in accordance with the ICP-MS analysis (W: 79.1 wt%).

### Characterization
XRD pattern was recorded on a Bruker AXS D8 Advance X-ray diffractometer with a Cu Ka radiation target (40 V, 40 A). TEM characterization was performed using an FEI Tecnai G2F20 microscope. Atomic-level HAADF-STEM images and the corresponding STEM-EDS elemental maps were measured on an FEI Titan Themis Z 3.1 equipped with a SCOR spherical aberration corrector and a monochromator. The

probe convergence angle was 80 mrad, and camera length was 115 mm in the STEM mode. The loading mass of W atoms in $W/WO_2$ materials and the dissolved W species were measured by the ICP-OES measurements (Agilent ICP-OES730). O K-edge NEXAFS spectra was performed at the Catalysis and Surface Science End-station at the BL11U beamline in the National Synchrotron Radiation Laboratory (NSRL) in Hefei, China. W $L_3$-edge X-ray adsorption spectra including XANES and EXAFS profiles were collected at BL11B station at the Shanghai Synchrotron Radiation Facility (SSRF).

For NAP-XPS characterization, in situ NAP-XPS measurements were conducted on a SPECS NAP-XPS instrument, where the photon source is the monochromatic X-ray source of Al Kα (1486.6 eV), and the overall spectra resolution is Ag 3d5/2, <0.5 eV FWHM at 20 kcps@UHV. The dried sample was transferred to the analysis chamber of XPS, and then W $4f$, C $1s$, O $1s$ XPS profiles were simultaneously recorded under the pressure of $1 \times 10^{-9}$ mbar. After introducing water molecules, the corresponding XPS signals were collected under the water atmosphere with a pressure of $1 \times 10^{-1}$ mbar.

For TOF-SIMS characterization, TOF-SIMS measurement was performed on a TOF-SIMS 5–100 instrument (ION-TOF GmbH) using a 30-keV $Bi^{3+}$ as analysis beam for negative and positive polarity measurements. The analysis beam current, raster size, and incident angles of all beam are 0.7 pA, $50 \times 50 \, \mu m^2$, and 45°, respectively.

For REELS characterization, the electrochemically treated materials were dried, and then transferred to the analysis chamber for REELS characterization using ESCALAB Xi equipment (Thermo Scientific). In detail, approximately 1000 eV electrons were incident on the surface of samples, the XPS analyzer would investigate the elastic and non-elastic scattering, corresponded to the elastic signal and energy loss peak (H loss). In our measurements, all parameters, such as the total acquisition time, energy step size, number of energy steps, and number of scans were set to be 2 mins, 0.02 eV, 1001, and 4, respectively.

For NMR characterization, $^1H$ Magic angle spinning nuclear magnetic resonance (MAS NMR) spectra was collected on a Bruker AVANCE NEO 400 MHz Wide Bore spectrometer operating at a magnetic field of 9.40 T. The chemical shifts for $^1H$ MAS NMR spectra was referenced to tetramethy (TMS). $^1H$ MAS NMR spectra was acquired at a spinning rate of 15 kHz with a π/2 pulse width of 3.5 μs and s recycle of 5 s.

For Py-IR characterization, Py-IR measurements were performed on a Nicolet 380 FT-IR instrument (Thermo Co., USA). The catalyst sample was pretreated at 353 K for 2 h under a vacuum condition, and background spectra were recorded in the ranging of 1700–1400 $cm^{-1}$ before adsorbing pyridine molecules. After reaching the adsorption equilibrium, the catalyst sample was retreated at 353 K for 4 h under a vacuum condition for the complete remove of physically adsorbed molecules, and the Py-IR spectra of pyridine chemisorption was obtained.

### Evaluation of HER performance

Electrochemical measurements were conducted on a CHI760E electrochemical station (Shanghai Chenhua Co., China) using a standard three-electrode system in 1 M KOH electrolyte, where free-standing $W/WO_2$ electrode (0.5 cm × 1.0 cm, area immersed in electrolyte is 0.5 cm × 0.5 cm), Ag/AgCl electrode, and a carbon rod were used as working, reference, and counter electrodes, respectively. For the fabrication of free-standing electrode for the physical mixture of W and $WO_2$ powders (tungsten loading mass: 3.2 $mg/cm^2$), W (5 mg) and $WO_2$ (5 mg) were dispersed into in a mixture of 800 μL ethanol, 170 μL water, and 30 μL nafion, followed by sonication treatment at least 30 min for preparation of the homogeneous catalyst ink. Then, 200 μL of the mixture catalyst ink was dropped onto the pre-treated Ni and pressed into a thinner foil for the following electrocatalytic measurement. For the HER measurement of commercial PtRu/C catalyst, 3.2 mg of PtRu/C powder was dispersed in a mixture of of 800 μL

ethanol, 170 μL water, and 30 μL Nafion, followed by sonication for at least 30 min to form a homogeneous catalyst ink. Then, all the catalyst ink was dropped onto the pre-treated Ni foam and pressed into a thinner foil for further measurement. For the preparation of W, $WO_2$, $W/WO_2$ catalyst ink, 5 mg of the powder materials is dispersed in the mixture of 800 μL ethanol, 170 μL water, and 30 μL nafion, followed by sonication treatment at least 30 min for the preparation of homogeneous catalyst ink. Then, 20 μL of the catalyst ink was dropped on the polished glassy carbon rotating disk electrode for electrocatalytic measurements at 1600 rpm. All potentials were recorded against the Ag/AgCl electrode and calibrated with respect to a reversible hydrogen electrode (RHE) in accordance with the Eq. (1):

$$E_{RHE} = E_{Ag/AgCl} + 1.02V \tag{1}$$

### TOF calculations

The $H_2$ conversion efficiencies of W, $WO_2$, and $W/WO_2$ can be evaluated from the TOF values, which were obtained according to the following Eq. (2):

$$TOF(s^{-1}) = \frac{j}{2Fn} \tag{2}$$

where $j$ (A) is the current at a given overpotential, 2 is the number of electrons consumed to form 1 mol $H_2$, F represents the Faraday constant (96500 C/mol), n (mol) is the number of moles of loaded metals, which can be evaluated based on the analysis of ICP-OES measurements (W wt%: W (80.8%), $WO_2$ (78.7%), $W/WO_2$ (79.1%))。

### DFT calculations

All computations were performed by applying the plane-wave-based DFT method as implemented in the Vienna Ab Initio Simulation Package (VASP) and periodic slab models. The projector augmented wave potentials with the Perdew–Burke–Ernzerhof form of the exchange-correlation functional were employed in all the simulations with the energy cut-off of 450 eV, and the long-range van der Waals (vdW) interaction was described with the DFT-D3 method. The $k$-point meshes were generated using the VASPKIT tool with the grid separation of 0.04 $Å^{-1}$ for the geometry optimizations and self-consistent field energy calculations, in which the total energy convergence and interaction force were set to be $10^{-6}$ eV and $10^{-2}$ eV/Å, respectively.

Based on the experimental TEM analysis, the structures of W and $WO_2$ substrates were built by cleaving the (110) plane of bulk W and (01-1) bulk $WO_2$, respectively. A vacuum region of 15 Å was set along the z direction to avoid the interaction between periodic images. Since the lattice parameters of $WO_2$ (01-1) plane were 7.38 Å and 5.58 Å, a 2 × 2 supercell was built as the $WO_2$ substrate for constructing the $W/WO_2$ interface. The (110) plane of W with the lattice parameter of 2.70 Å was enlarged to build a 3 × 4 supercell, which was then cleaved (100) surface to build a W slab whose (110) and (110) surfaces were exposed. The $W/WO_2$ interface was constructed by placing the W slab on the $WO_2$ substrate. The lattice mismatch between W slab (10.8 Å) and $WO_2$ substrate (11.2 Å) was only about 3%. The structures of constructed $W/WO_2$ interface with front, side, and top images can be observed in Supplementary Fig. 21.

For evaluating the $H_2O$ dissociation energy barrier, the transitional state was located using the Nudged Elastic Band method. The free energy of adsorbed H ($\Delta G_H$) on surfaces is expressed as Eq. (3):

$$\Delta G_H = \Delta E_H + \Delta E_{ZPE} - T\Delta S \tag{3}$$

where $\Delta E_H$ is the hydrogen adsorption energy, $\Delta E_{ZPE}$ and $\Delta S$ are the zero point energy difference and the entropy difference between the

adsorbed state and the gas phase, respectively, and T is the system temperature (298.15 K).

## Data availability

Data reported herein have been deposited in the Figshare database, and are accessible through https://doi.org/10.6084/m9.figshare.23578770. Source data are provided with this paper.

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

## Acknowledgements

Y.C. would like to acknowledge the supported from the National Key R&D Program of China (Nos. 2022YFA1503801 and 2022YFA1503802), the National Natural Science Foundation of China (22172190), the CAS Project for Young Scientists in Basic Research (No. YSBR-022), and the Young Cross Team Project of CAS (No. JCTD–2021-14). Z.C. would like to acknowledge the support from the National Natural Science Foundation of China (No. 22109171). W.G. would like to acknowledge the support from the National Natural Science Foundation of China (No. 12004324), Science and Technology Project of Jiangxi Province (No. 20202ACBL211005), and Natural Science Foundation of the Jiangsu Higher Education Institutions of China (No. 22KJA140003). The authors are grateful for the technical support of Nano-X from Suzhou Institute of Nano-Tech and Nano-Bionics, Chinese Academy of Sciences (SINANO), in particular, we thank R.H. and Z.L. for their help in the measurements and analysis of REELS and TOF-SIMS, respectively. We also thank Q.X., J.H., H.D., and J.Z. for their help in Near Edge X-ray Absorption Fine Structure (O K-edge NEXAFS) performed at the Catalysis and Surface Science End-station at the BL11U beamline in the National Synchrotron Radiation Laboratory (NSRL) in Hefei, China. We thank J.W. and R.G. for their help in the measurements and analysis of X-ray adsorption spectra (W-$L_3$ edge XAS) characterization at BL11B station in the Shanghai Synchrotron Radiation Facility (SSRF).

## Author contributions

Y.C. conceived the project and designed the experiments. Z.C., S.H., G.Y., C.Z., X.F., and Y.L. performed the synthesis and characterization of catalysts and the electrocatalytic measurements. Z.C. R.G., and J.W. performed the X-ray adsorption experiments and analyzed the raw data. Z.C. and W.G. carried out the DFT calculations. Z.C. and Y.C. wrote the paper. All authors discussed the results and commented on the manuscript.

## Competing interests

The authors declare no competing interests.
