## [Peer Review File · Nature Communications]

Metallic W/WO₂ solid-acid catalyst boosts hydrogen evolution reaction in alkaline electrolyteREVIEWER COMMENTS

Reviewer #1 (Remarks to the Author):

Chen et al. presented an interesting study where W/WO₂ catalyst was produced and demonstrated as highly effective for H₂ evolution in alkaline media. The experimental side of the work is very detailed and presents a concerted use of different advanced characterization techniques intended to unveil the mechanism of high HER activity of W/WO₂ catalysts. Moreover, the experimental findings were supplemented by DFT calculations. The presented picture is plausible but lacks an explanation of how dissolution was prevented. Also, there are numerous other points that should be resolved, as indicated below, before considering the present work for publication in Nature Communications.

- Language can be improved to make the manuscript easier to read
- Authors mentioned "appealing electronic interaction between zero-valence W atoms and protons" citing the work which actually refers to WC as H₂ evolution catalyst. It is accepted that strong metallic W-H₂ interactions render poor HER activity of tungsten. Could Authors be more specific about their claims?
- Looking at the Pourbaix plot of tungsten, at high pH, metallic W actively dissolves. The Authors also mentioned that their catalyst is less prone to dissolution, so a detailed justification of the twist of W dissolution behavior is needed.
- What is the role of carbon and the carbon sources in the W/WO₂ synthesis? Please make sure to clearly present your strategy from the very beginning.
- Why PtRu/C was selected as a benchmark?
- Please re-do the Tafel analysis at potentials that are far enough from 0 V vs. RHE (zero overvoltage). As a note, HER is the cathodic reaction, thus, current densities, overvoltages, and Tafel slopes should all be negative.
- EIS is always in operando, it is not possible to do it ex-situ.
- Could Authors show EIS at lower frequencies (minimum was 0.1 Hz) and discriminate between capacitance or some additional resistance components? Also, the discussion is very descriptive, without proper quantification of experimental data. Which circuits were used to model EIS and extract discussed parameters?
- Stability is not actually very good, but I was also wondering why electrolyte was replaced for the measurements at -50 mA cm⁻²? Also, are there any tungsten species in the solution, suggesting dissolution?
- HER polarization curves go to relatively high currents, but stability was tested at lower currents. Why not providing stability tests at higher currents (at least 200 mA cm⁻², which is the lower limit for industrial alkaline electrolysis)?
- Going back to the poor HER activity of W, what is the barrier for the formation of H₂ from 2H_{ads} adsorbed on W, and how this relates to the Authors' findings?
- Much more detail are needed for the description of DFT calculations. As presented now, the calculations cannot be reproduced.

Reviewer #2 (Remarks to the Author):

Electrocatalytic hydrogen evolution is an important technology to convert renewable electricity into hydrogen fuel. Developing highly efficient, cheap HER catalysts are key to make this technology widely useful. In this work, the authors have synthesized a W/WO₂ catalyst which shows decent performance in alkaline electrolytes, demonstrate an excellent water dissociation performance. The work is

interesting, cleanly done, and well organized. I would like to recommend its publication in Nature Communications after the following minor revisions.

1. In the step of thermal decomposition precursor, is it possible for C to infiltrate into the material lattice and form low-crystallinity WC or W₂C? Please demonstrate that there is no infiltration of C during the synthesis of W/WO₂, and prove the existence of graphite carbon in the material through the d/g peak in Raman spectra.
2. By comparing Figure 4c and Figure S14, it can be observed that both pure-phase W and WO₂ do not have the signals of ads. H₂O. Please explain this phenomenon.
3. For Figure 4h, it can be seen that with the increase of applied potential, the proportion of W-OH T/W-OH L decreases. Please analyze this phenomenon.
4. In the DFT calculation of water dissociation, some literatures have reported that water adsorbs on the material surface to produce H₂O*, and the free energy of this step is not 0 eV. Please confirm whether the results in Figure 5b need to be corrected.
5. The authors used methods such as REELS, NMR, Py-IR, and NAP-XPS in the experiment. Please supplement detailed test steps in the "Method" section.

Reviewer #3 (Remarks to the Author):

The authors synthesized a heterostructure composed of tungsten nanoparticles and WO₂ crystals, acting as a hydrogen evolution reaction (HER) catalyst on nickel foam. The catalyst was prepared by calcinating a mixture of W₁₈O₄₉ nanowires and an organic carbon source. For comparative purposes, the authors also prepared bare W and WO₂ catalysts for the HER. The W/WO₂ catalyst demonstrated significantly superior catalytic performance for the HER in alkaline conditions compared to the other two catalysts. Its enhanced activity level was roughly equivalent to that of a noble metal catalyst. Notably, the current density was consistently maintained for 50 hours. This research intriguingly suggests the potential of utilizing low-cost metal oxides as HER catalysts in alkaline conditions. However, the characterization and illustration is not enough to describe the structural properties of W/WO₂ heterostructure, and additional demonstration is required to support the author's statement. Thus, this manuscript should be revised before publication in the Nature Communications.

Comments are attached below:

1. There is a discrepancy between the terminology used within the manuscript and that used in the accompanying figures when referring to the prepared tungsten nanoparticles and WO₂ nanorods. For instance, the term 'phase-pure WO₂' is utilized in the manuscript, whereas 'bare WO₂' appears in the figures (for example, Figure 2b or 2c). For consistency, it is recommended to revise the terminology across both the manuscript and the figures.
2. The W/WO₂ catalyst was calcined using a 5% H₂/Ar and at 750 °C, and carbon residue was visible in the TEM images (Figure 1 and S4). Could there be a possibility of tungsten carbide (WC) synthesis? WC is typically studied as another type of HER catalyst. To conclude that improved catalytic performance is originated from the W/WO₂, evidence is needed to exclude the presence of WC. C 1s XPS spectra would help confirm that the influence of any potential WC catalyst on the HER can be disregarded.
3. The authors attributed the enhanced catalytic performance of the W/WO₂ heterostructure to the abundance of oxygen vacancies, as revealed by ESR and NEXAFS analyses at the O K edge. However, the XPS O 1s spectra, which also contain the information about the oxygen vacancies, were not presented in this manuscript. XPS results could provide additional support by confirming the quantity of oxygen vacancies formed within the prepared catalyst.
4. In Figure S11, a new NEXAFS peak emerges for the used W/WO₂ at 540 eV. Commonly, an O K

edge NEXAFS spectra peak around 540 eV is indicative of interaction between the O 2s orbital and the metal sp state. Please provide further explanation about the significance and implications of this new peak.

5. The illustration of Figure 4e and f was omitted in the manuscript and caption. Please describe the meaning of that figure.

6. What about assessing the catalytic activity of the W, WO₂, and W/WO₂ catalysts using the Rotating Disk Electrode (RDE) technique? Evaluating the HER activity of the prepared catalysts through RDE measurements could provide a more quantitative comparison of their catalytic activity relative to previously reported catalysts.

7. The authors computed the transition energy for the water dissociation pathway on W, WO₂, and at the W/WO₂ interface, with the corresponding simulated models displayed in Figures 5 and S18. Could the authors specify the bond length between the oxygen and dissociated hydrogen atom after dissociation? It appears that this bond length on the W/WO₂ interface may be shorter than in the other two cases, which might contribute to the smaller energy barrier. Additionally, more comprehensive information, such as top and side images of the calculated structure, would be beneficial to better illustrate the W/WO₂ interface.

8. The author insisted that heterojunction between the W nanoparticle and WO₂ structure is the reason of the improved catalytic activity. How about the HER activity of the catalyst that synthesized by physical mixing of the W nanoparticle and WO₂ nanorod?

Answer to the comments of reviewers:

Replies on comments of reviewer 1:

Chen et al. presented an interesting study where W/WO₂ catalyst was produced and demonstrated as highly effective for H₂ evolution in alkaline media. The experimental side of the work is very detailed and presents a concerted use of different advanced characterization techniques intended to unveil the mechanism of high HER activity of W/WO₂ catalysts. Moreover, the experimental findings were supplemented by DFT calculations. The presented picture is plausible but lacks an explanation of how dissolution was prevented. Also, there are numerous other points that should be resolved, as indicated below, before considering the present work for publication in Nature Communications.

Author reply: Thanks for the referee's valuable comments. We have supplemented additional experiments and demonstrations to address the referee's concerns, in particular, detailed characterizations have been performed to better understand the twist of tungsten dissolution for W/WO₂ materials in high-pH solutions.

Comments 1:

Language can be improved to make the manuscript easier to read.

Author reply: Thanks for the referee's valuable comment. The language of this manuscript has been polished thoroughly.

Comments 2:

Authors mentioned "appealing electronic interaction between zero-valence W atoms and protons" citing the work which actually refers to WC as H₂ evolution catalyst. It is accepted that strong metallic W-H_{ads} interactions render poor HER activity of tungsten. Could authors be more specific about their claims?

Author reply: Thanks for the referee's careful reading and professional comments. Generally, our cited work demonstrates that the introduction of metallic W into WC materials can efficiently regulate the hydrogen adsorption energies on catalyst surface when the produced hydrogen

intermediates locate at the bridging and/or hollow sites between zero-valence W atoms in alkaline HER process (*Adv. Sci.* **9**, 2106029 (2022)). In our work, we mainly would like to illustrate that the high-valence W^{4+} oxidation species (WO_2) benefits the construction of Brønsted acid sites for achieving a proton-concentrated catalyst surface, while the interfacial zero-valence W (W^0) sites prefer to accelerate the deprotonation kinetics of neighboring Brønsted acid sites. The significant advantage of interfacial W^0 sites for hydrogen desorption is predicted by the subsequent DFT calculations (Fig. 5c). In addition, for the viewpoint that strong metallic $W-H_{ads}$ interactions render poor HER activity of tungsten, we would like to point out that the metallic $W-H_{ads}$ interactions are largely depended on the chemical and electronic environments of metallic W atoms (Fig. 5c), bare W metal exhibits a relatively strong $W-H_{ads}$ interaction with the ΔG_H up to -0.51 eV, while the value is substantially decreased (-0.41 eV) when H intermediates are located at the interfacial W^0 atoms (Fig. 5c), and can be further optimized with the increase of proton coverage (Brønsted acid sites) at the W/WO_2 interface (Fig. 5e). Accordingly, the more specific text has been added in the revised manuscript. “(iii) considering the relatively sluggish hydrogen desorption kinetics of H_xWO_y intermediates, the introduction of zero-valence W (W^0) sites can further accelerate the deprotonation kinetics of Brønsted acids for the cycling of active sites due to the optimized electronic interactions between W^0 atoms and protons at the W/WO_2 interface.”

Fig. 5 (c) The calculated free energy diagrams of hydrogen desorption step on W, WO_2 , and W/WO_2 catalyst surface. (e) Free energies of hydrogen desorption from W/WO_2 interface tuned by continuously increased neighboring Brønsted acid sites.

Comments 3:

Looking at the Pourbaix plot of tungsten, at high pH, metallic W actively dissolves. The authors also mentioned that their catalyst is less prone to dissolution, so a detailed justification of the twist of W dissolution behavior is needed.

Author reply: Thanks for the referee's constructive suggestion. Prior to discuss the dissolution behaviors of the as-prepared three types of tungsten-based materials, we would like to point out that all tungsten-based materials are supported and/or encapsulated by graphite carbon layers (synthetic strategy and morphology characterizations), thus surface carbon layers cannot be neglected for the protection of inner tungsten species (*Adv. Mater.* **34**, 2202743 (2022); *Nano Energy* **69**, 104455 (2020)). In order to better illustrate the advantage of the as-prepared W/WO₂ in anti-dissolution property under high-pH solutions, we have detected the concentrations of the dissolved W species for W, WO₂, and W/WO₂ powders in 1 M KOH solutions. In detail, W (1 mg), WO₂ (1 mg), and W/WO₂ (1 mg) were added to 5 mL KOH (1 mol/L) solutions, respectively. After 20 mins, the concentration of dissolved W species was determined by inductively coupled plasma-optical emission spectroscopy (ICP-OES, Agilent ICP-OES730), and the values of dissolved W species are 7.1×10^{-4} , 1.1×10^{-4} , and 6.3×10^{-5} mol/L for W, WO₂, and W/WO₂, respectively. Compared to WO₂ and W/WO₂ materials, W powder exhibits a much higher concentration of dissolved W species, probably because W nanoparticles with sizes less than 5 nm have poor oxidation resistance in alkaline solutions. After 2 days, the detected concentration of dissolved W is 3.4×10^{-4} and 9.6×10^{-5} mol/L for WO₂ and W/WO₂ samples, respectively, while the value is as high as 4.7×10^{-3} for W nanoparticles, respectively, suggesting the long-term alkaline leaching has severely caused the dissolution of W powder. Based on above experimental evidences, two significant points can be concluded: (i) WO₂ and W/WO₂ heterostructures have relatively strong anti-dissolution property in high-pH solutions, which may be attributed to their intrinsically good oxidation resistance with co-existence of metal and oxide features (*J. Mater. Chem. C*, **6**, 3200-3205 (2018); *Adv. Energy Mater.* **12**, 2103301 (2022)) and the protection of surface carbon layers, meanwhile, the strong chemical and electronic interactions of W and WO₂ components within W/WO₂ may further improve the structural robustness in alkaline solutions (*ACS Catal.* **10**, 13227-13235 (2020); *Mater. Today Nano* **6**, 100038 (2019)); (ii) the ultra-small W NPs have poor alkaline leaching resistance, but we cannot observe the complete dissolution of W powders due to the protection of carbon layers. In addition, we would like to point out that partial tungsten species of initial ICP-OES detection may be originated from the naturally dissolution of inevitable high-valence tungsten oxides on low-valence WO₂ and W/WO₂ samples (*Adv. Energy*

Mater. **12**, 2103301 (2022); *J. Am. Chem. Soc.* **139**, 5285-5288 (2017); *Angew. Chem. Int. Ed.* **53**, 5131-5136 (2014)).

Figure. The investigation of anti-dissolution property of W/WO₂ in comparison with W and WO₂ counterparts.

Comments 4:

What is the role of carbon and the carbon sources in the W/WO₂ synthesis? Please make sure to clearly present your strategy from the very beginning.

Author reply: Thanks for the referee's valuable comment. The carbon sources are organic polyethylene oxide-co-polypropylene oxide-co-polyethylene oxide (P₁₂₃) and dopamine (DA). In the pyrolysis process, the pyrolyzed carbon can cause the reduction reaction ($C+W_{18}O_{49}\rightarrow W+WO_2+CO_2\uparrow$) of high-valence W₁₈O₄₉ parent materials for the synthesis of desired W and/or WO₂ products, meanwhile, the produced graphite carbon layers not only improve the electron transfer in following alkaline HER process, but also alleviate the alkaline-leaching rate of inner W/WO₂ materials. Accordingly, the following text has been added in the revised manuscript. "where the whole reduction reactions can be simplified as following equations: $C+W_{18}O_{49}\rightarrow W+WO_2+CO_2\uparrow$, $H_2+W_{18}O_{49}\rightarrow W+WO_2+H_2O\uparrow$. In the pyrolysis process, carbon can cause the reduction process of high-valence W₁₈O₄₉ parent materials for the synthesis of desired W and WO₂ products. In the following alkaline HER process, the produced graphite carbon layers not only improve the electrocatalytic charge transfer, but also alleviate the alkaline-leaching rate of inner W/WO₂ materials."

Comments 5:

Why PtRu/C was selected as a benchmark?

Author reply: Thanks for the referee's valuable comment. Pt-based alloying catalysts have been widely used in alkaline HER process (*J. Am. Chem. Soc.* **140**, 9046-9050 (2018); *Nat. Commun.* **8**,

14580 (2017)), because the heteronuclear metal sites are generally efficient in either breaking the H-OH bonds, or further optimizing the Pt-H* binding energies (*ACS Catal.* **9**, 10870-10875 (2019)), both are urgently demanded to be optimized in pure Pt catalysts for alkaline HER process (*Angew. Chem. Int. Ed.* **58**, 13107-13112 (2019); *Nano Energy* **81**, 105636 (2021)). Among numerous Pt-based alloying catalysts, noble metal PtRu/C (Pt₁Ru₁, 20 wt%) is commercialized, and represents an ideal HER alloying catalyst with optimized reaction kinetics of water dissociation and hydrogen desorption in alkaline electrolyte (*Angew. Chem. Int. Ed.* **60**, 5771-5777 (2021)). In our work, from the viewpoint of constructing solid-acid catalyst surface using tungsten-based materials, we mainly declare that the energy barriers of water dissociation and hydrogen desorption steps can be substantially decreased on W/WO₂ heterostructure catalyst in alkaline HER process. Accordingly, commercial PtRu/C is selected as the benchmark HER catalyst in our work.

Comments 6:

Please re-do the Tafel analysis at potentials that are far enough from 0 V vs. RHE (zero overvoltage). As a note, HER is the cathodic reaction, thus, current densities, overvoltages, and Tafel slopes should all be negative.

Author reply: Thanks for the referee's valuable comment. As suggested by the referee, we have re-done the Tafel analysis with regions near 0 V (Fig. 3d). As can be seen, the current responses are extremely weak for all catalysts when overpotentials are positive (>0 V), and such weak signals may be originated from the capacitance of naturally adsorbed electrolyte on Ni-foam electrode (0.5 cm × 0.5 cm, Ni foam). However, slow current responses are observed on all large-area free-standing catalysts (slow begin of HER process on large-area electrode) when overpotentials just entering to negative regions (<0 V). Afterwards, the alkaline HER process starts to enter the Tafel region with the fastest reaction rate. Similar Tafel regions are also observed in other free-standing systems (*Nat. Commun.* **12**, 6776 (2021); *Nat. Commun.* **9**, 924 (2018); *Adv. Mater.* **31**, 1904989 (2019)). In addition, we have also tried to eliminate the influence of slow begin of HER process for the selection of Tafel regions by using rotating disk electrode (RDE) technique with glassy carbon working electrode (diameter: 3 mm) (measurement details seen in the method section), where current responses are sensitive and smooth when overpotentials entering to negative potential regions. W/WO₂ catalyst still exhibits a remarkable alkaline HER

activity with a low overpotential (-60 mV) at -10 mA/cm² and a small Tafel slope (-54 mV/dec) (Supplementary Fig. 9), which are more excellent than those of W and WO₂ counterparts, and still excelling most previously reported metal oxides (Fig. 3c). Accordingly, the following text has been added in the revised manuscript. “Moreover, the HER performance of W, WO₂, W/WO₂ powders were also examined using the rotating disk electrode (RDE) technique in 1 M KOH electrolyte. As can be seen, W/WO₂ catalyst still exhibits a remarkable alkaline HER activity with a low overpotential (-60 mV) at -10 mA/cm² and a small Tafel slope (-54 mV/dec) (Supplementary Fig. 9), which are more excellent than those of W and WO₂ counterparts, and still excelling most previously reported metal oxides (Fig. 3c).”

Fig 3. (a) The raw data including the capacitance of naturally adsorbed electrolyte, slow begin of HER process, and Tafel regions for all catalysts. (b) Tafel plots of all catalysts extracted from (a). (c) Polarization curves and (d) the corresponding Tafel plots of W, WO₂, W/WO₂ catalysts examined by RDE technique.

Comments 7:

EIS is always in operando, it is not possible to do it ex-situ.

Author reply: Thanks for the referee’s valuable comment. All inappropriate descriptions have been corrected in the revised manuscript.

Comments 8:

Could authors show EIS at lower frequencies (minimum was 0.1 Hz) and discriminate between capacitance or some additional resistance components? Also, the discussion is very descriptive, without proper quantification of experimental data. Which circuits were used to model EIS and extract discussed parameters?

Author reply: Thanks for the referee's valuable comments. Generally, all presented EIS curves were collected in the frequency range of 10^{-1} ~ 10^5 Hz (Fig. 3f), which can be confirmed by the corresponding Bode phase plots in Fig. 3g. All Nyquist plots of W/WO₂ samples were simulated by a double-parallel equivalent circuit model in accordance with previous reports (*Energy Environ. Sci.*, **12**, 2298-2304 (2019); *J. Power Sources*, **158**, 464-476 (2006)), where R_1 represents the uncompensated solution resistance, the first parallel components (constant phase element (CPE) and R_2) indicate the charge transfer resistance caused by the adsorption and activation of water molecules at low frequencies (*Energy Environ. Sci.*, **14**, 6428-6440 (2021); *ACS Appl. Energy Mater.* **3**, 66-98 (2020)), and the second parallel ones of R_2 and C_ϕ are attributed to the hydrogen adsorption resistance and pseudo-capacitance at high frequencies, respectively. As expected, all electrochemically treated W/WO₂ samples exhibit the similar R_1 value (~4.0 Ω), and the small values of R_2 for all W/WO₂ catalysts suggest the fast charge transfer kinetics between catalyst surface and H₂O molecules. In particular, R_2 decreases to 4.4 Ω sharply with a negligible water diffusion resistance at applied overpotential of -20 mV, indicating the adsorption and activation of water molecules on the W/WO₂ catalyst surface can be achieved under low overpotentials. Further, we also notice that R_3 and C_ϕ are largely overpotential-dependent, where W/WO₂ catalysts exhibit significantly decreased R_3 with increased C_ϕ when increasing the applied overpotentials, in particular, the value of R_3 can be as low as approximately 1.8 Ω , while C_ϕ is up to 0.017 F at an overpotential of -30 mV, suggesting the hydrogen adsorption resistance (R_3) is very small, and the capacitance of proton coverage is very large on W/WO₂ catalyst surface under low overpotentials.

Accordingly, the relevant discussions have been added in the revised manuscript. "All Nyquist plots of W/WO₂ samples were simulated by a double-parallel equivalent circuit model in accordance with previous reports (*Energy Environ. Sci.*, **12**, 2298-2304 (2019); *J. Power Sources*, **158**, 464-476 (2006)), where R_1 represents the uncompensated solution resistance, the first parallel components (constant phase element (CPE) and R_2) indicate the charge transfer resistance caused

by the adsorption and activation of water molecules at low frequencies (*Energy Environ. Sci.*, **14**, 6428-6440 (2021); *ACS Appl. Energy Mater.* **3**, 66-98 (2020)), and the second parallel ones of R_2 and C_ϕ are attributed to the hydrogen adsorption resistance and pseudo-capacitance at high frequencies, respectively (Fig. 3f and Supplementary Table 1). As expected, all electrochemically treated W/WO₂ samples exhibit the similar R_1 value ($\sim 4.0 \Omega$), and the small values of R_2 for all W/WO₂ catalysts suggest the fast charge transfer kinetics between catalyst surface and H₂O molecules. In particular, R_2 decreases to 4.4Ω sharply with a negligible water diffusion resistance at applied overpotential of -20 mV, indicating the adsorption and activation of water molecules on the W/WO₂ catalyst surface can be achieved under low overpotentials. Further, we also notice that R_3 and C_ϕ are largely overpotential-dependent, where W/WO₂ catalysts exhibit significantly decreased R_3 with increased C_ϕ when increasing the applied overpotentials, in particular, the value of R_3 can be as low as approximately 1.8Ω , while C_ϕ is up to 0.017 F at an overpotential of -30 mV, suggesting the hydrogen adsorption resistance (R_3) is very small, and the capacitance of proton coverage is very large on W/WO₂ catalyst surface under low overpotentials.”

Fig. 3 (f) Nyquist plots of W/WO₂ catalysts with the increase of applied overpotentials, the inset shows the equivalent circuit for the simulation. Note that inhomogeneities in the surface of metal oxide electrodes usually result in non-ideal capacitance in the double-layer at the solid/electrolyte interface. Thus, CPEs (CPE-T and CPE-P) are routinely used in place of pure capacitors to model this interfacial layer.

Comments 9:

Stability is not actually very good, but I was also wondering why electrolyte was replaced for the measurements at -50 mA cm^{-2} ? Also, are there any tungsten species in the solution, suggesting dissolution?

Author reply: Thanks for the referee's valuable comments. The larger current density means the higher consumption of water molecules, and the replacement of electrolyte can guarantee the supply of HER reactant, which meanwhile prevents the W/WO₂ catalyst from the possible alkaline leaching by the continuously concentrated OH⁻ intermediates near cathodic electrode after long-term hydrogen production. The operations of refreshing electrolyte have been widely used in electrolysis (*Angew. Chem. Int. Ed.* **62**, e202300390 (2023); *Energy Environ. Sci.*, **13**, 119-126 (2020); *Adv. Funct. Mater.* **31**, 2010367 (2021)), and the activity can be recovered when refreshing the electrolyte in our work, suggesting the relatively stable HER process using W/WO₂ solid-acid catalyst. On the other hand, the larger current density also means the higher hydrogen production, and the release of H₂ gas bubbles also can cause the fluctuation of the collected stability curves (*Nat. Commun.* **12**, 3540 (2021); *Nat. Commun.* **9**, 924 (2018); *J. Mater. Chem. A*, **10**, 6242-6250 (2022)). However, the overpotentials at -10 and -50 mA/cm² of the Chronopotentiometry measurements are still maintained near the values of -35 mV and -115 mV in our work, which also suggests the good stability performance of W/WO₂ solid-acid catalyst in alkaline HER process.

In addition, similar to previous report (*Nat. Commun.* **9**, 2609 (2018)), inductively coupled plasma-optical emission spectroscopy (ICP-OES) was performed to detect the possibly dissolved tungsten species of W/WO₂ catalyst in the used KOH electrolyte (50 mL, 1 mol/L KOH). As can be seen, the concentration of W is determined to be approximately 7.5×10^{-6} mol/L when the catalyst electrode is immersed in KOH electrolyte under open circuit potential condition, partial of which can be attributed to the dissolution of inevitable high-valence tungsten oxides (e.g., WO₃) on W/WO₂ catalyst surface. After long-term hydrogen production, the detected dissolved W species in the used and refreshed electrolytes (indicated by the red dashed circle, Fig. 3i) are approximately 9.8×10^{-6} and 6.3×10^{-7} mol/L (refreshed electrolyte), respectively, suggesting the extremely slow dissolution of W/WO₂ catalyst in alkaline HER process. Meanwhile, unlike the naturally alkaline leaching under circuit potential condition, we would like to point out that the negative potentials at cathode can provide rich electrons to avoid the oxidation and dissolution of low-valence tungsten species during alkaline HER process, which causes rather low concentration (6.3×10^{-7} mol/L) of dissolved tungsten species detected in the refreshed electrolyte, suggesting

structural robustness of W/WO₂ catalyst after long-term hydrogen production in alkaline electrolyte.

Therefore, combining the detailed characterizations in **Comments 3** of Reviewer 1, the significantly improved anti-dissolution property of our well-designed W/WO₂ composites in high-pH electrolyte can be understood by the following two reasons: (i) W/WO₂ heterostructures have intrinsically good oxidation resistance with co-existence of metal and oxide features (*J. Mater. Chem. C*, **6**, 3200-3205 (2018); *Adv. Energy Mater.* **12**, 2103301 (2022)) and the protection of surface carbon layers (*Adv. Mater.* **34**, 2202743 (2022); *Nano Energy* **69**, 104455 (2020)), meanwhile, the strong chemical and electronic interactions of W and WO₂ components within W/WO₂ may further improve the structural robustness in alkaline solutions (*ACS Catal.* **10**, 13227-13235 (2020); *Mater. Today Nano* **6**, 100038 (2019)); (ii) unlike the naturally alkaline leaching under circuit potential condition, the negative potentials at cathode can provide rich electrons to avoid the oxidation and dissolution of low-valence tungsten species during alkaline HER process.

Accordingly, the relevant discussions have been added in the revised manuscript.

“The relatively good stability of W/WO₂ solid-acid catalyst in alkaline electrolyte is also confirmed by the inductively coupled plasma-optical emission spectroscopy (ICP-OES). After long-term hydrogen production, the initial electrolyte, used electrolyte, and refreshed electrolyte (indicated by red dashed circle, Fig. 3i) shows no significant increase in the concentrations of dissolved W species with values of 7.5×10^{-6} , 9.8×10^{-6} , and 6.3×10^{-7} mol/L, respectively. In particular, the rather low concentration (6.3×10^{-7} mol/L) of dissolved W species in the refreshed electrolyte directly suggests the structural robustness of W/WO₂ catalyst after long-term hydrogen production in alkaline electrolyte.”

“Therefore, based on the comprehensive evaluations of alkaline HER activity and stability on W/WO₂ materials, the significantly improved structural robustness of our well-designed W/WO₂ composites in high-pH solutions can be attributed to the following two reasons: (i) W/WO₂ heterostructures have intrinsically good oxidation resistance with co-existence of metal and oxide features (*J. Mater. Chem. C*, **6**, 3200-3205 (2018); *Adv. Energy Mater.* **12**, 2103301 (2022)) and the protection of surface carbon layers (*Adv. Mater.* **34**, 2202743 (2022); *Nano Energy* **69**, 104455 (2020)), meanwhile, the strong chemical and electronic interactions of W and WO₂ components

within W/WO₂ may further improve the structural robustness in alkaline solutions (*ACS Catal.* **10**, 13227-13235 (2020); *Mater. Today Nano* **6**, 100038 (2019)); (ii) unlike the naturally alkaline leaching under circuit potential condition, the negative potentials at cathode can provide rich electrons to avoid the oxidation and dissolution of low-valence tungsten species during alkaline HER process.”

Comments 10:

HER polarization curves go to relatively high currents, but stability was tested at lower currents. Why not providing stability tests at higher currents (at least 200 mA cm⁻², which is the lower limit for industrial alkaline electrolysis)?

Author reply: Thanks for the referee’s valuable comment. According to the constructive suggestion of the referee, the electrocatalytic stability of W/WO₂ catalyst was evaluated in alkaline electrolyte at a current density of -200 mA/cm². Compared to the relatively stable profile collected at -50 mA/cm², although W/WO₂ catalyst exhibits much stronger fluctuation in activity at current density of -200 mA/cm², the activity can be recovered after refreshing the electrolyte, suggesting the good catalytic stability at industrial current density.

Accordingly, the following text has been added in the revised manuscript. “In addition, we also evaluate the stability of W/WO₂ catalyzing alkaline HER process at a current density of -200 mA/cm² (the lower limit for industrial water electrolysis). No significant activity loss can be observed on W/WO₂ catalysts after long-term hydrogen production, suggesting the good catalytic stability at industrial current density.”

Fig. 3 (i) Chronopotentiometry measurements of W/WO₂ solid-acid catalyst at current densities of 10, 50, and 200 mA/cm².

Comments 11:

Going back to the poor HER activity of W, what is the barrier for the formation of H_2 from $2H_{ads}$ adsorbed on W, and how this relates to the authors' findings?

Author reply: Thanks for the referee's valuable comments. According to the Tafel analysis, the alkaline HER process follows the Volmer-Heyrovsky reaction pathway on pure W catalyst surface (Fig. 3d), indicating the rate determining step of W catalyzing alkaline HER process is the water dissociation step, and the produced H_2 molecule originates from the combination reaction of one H_{ads} intermediate and one H_2O molecule ($H_{ads} + H_2O + e^- \rightarrow H_2\uparrow + OH^-$) (*Chem. Rev.* **120**, 851-918 (2020)). Meanwhile, the DFT calculations also give the consistent information, which suggests that pure W is generally inefficient in the cleavage of H-OH bonds with energy barrier up to 0.84 eV (Fig. 5b), whereas the energy barrier of hydrogen desorption can be substantially decreased to -0.51 eV (-0.51 eV). Conversely, metallic WO_2 exhibits an extremely low energy barrier (0.06 eV) in water dissociation, but is poor at resulting in the desorption (-1.24 eV) of adsorbed H^* intermediates. Therefore, an optimal catalyst can be designed by combining the catalytic proficiencies of W and WO_2 materials, where the introduction of W component can accelerates the desorption kinetics of protons on neighboring WO_2 catalyst surface.

Fig. 5 The calculated free energy diagrams of (b) water adsorption and dissociation, and (c) hydrogen desorption steps on W, WO_2 , and W/ WO_2 catalyst surface.

Comments 12:

Much more detail are needed for the description of DFT calculations. As presented now, the calculations cannot be reproduced.

Author reply: Thanks for the referee's valuable comment. More detailed information of DFT calculations has been provided for the reproduction of simulation results in the revised manuscript. Accordingly, the following text has been added in the method section for DFT calculations. "The

projector augmented wave potentials with the Perdew–Burke–Ernzerhof form of the exchange-correlation functional were employed in all the simulations with the energy cut-off of 450 eV, and the long-range van der Waals (vdW) interaction was described with the DFT-D3 method. The k -point meshes were generated using the VASPKIT tool with the grid separation of 0.04 \AA^{-1} for the geometry optimizations and self-consistent field energy calculations, in which the total energy convergence and interaction force were set to be 10^{-6} eV and 10^{-2} eV/Å, respectively.

Based on the experimental TEM analysis, the structures of W and WO₂ substrates were built by cleaving the (110) plane of bulk W and (01-1) bulk WO₂, respectively. A vacuum region of 15 Å was set along the z direction to avoid the interaction between periodic images. Since the lattice parameters of WO₂ (01-1) plane were 7.38 Å and 5.58 Å, a 2 x 2 supercell was built as the WO₂ substrate for constructing the W/WO₂ interface. The (110) plane of W with a lattice parameter of 2.70 Å was enlarged to build a 3 x 4 supercell, which was then cleaved (100) surface to build a W slab whose (110) and (110) surfaces were exposed. The W/WO₂ interface was constructed by placing the W slab on the WO₂ substrate. The lattice mismatch between W slab (10.8 Å) and WO₂ substrate (11.2 Å) was only about 3%. The structures of constructed W/WO₂ interface with front, side, and top images can be observed in Supplementary Fig. 21.”

Replies on comments of reviewer 2:

Electrocatalytic hydrogen evolution is an important technology to convert renewable electricity into hydrogen fuel. Developing highly efficient, cheap HER catalysts are key to make this technology widely useful. In this work, the authors have synthesized a W/WO₂ catalyst which shows decent performance in alkaline electrolytes, demonstrate an excellent water dissociation performance. The work is interesting, cleanly done, and well organized. I would like to recommend its publication in Nature Communications after the following minor revisions.

Author reply: We appreciate the reviewer’s comments and recommendation very much. We have supplemented additional experiments and demonstrations to address the reviewer’s concerns.

Comments 1:

In the step of thermal decomposition precursor, is it possible for C to infiltrate into the material lattice and form low-crystallinity WC or W₂C? Please demonstrate that there is no infiltration of C

during the synthesis of W/WO₂, and prove the existence of graphite carbon in the material through the D/G peak in Raman spectra.

Author reply: Thanks for the referee's constructive comments. X-ray photoelectron spectroscopy (XPS) is very sensitive to identify the surface carbon species in carbon supported active materials (*ACS Nano* **9**, 5125-5134 (2015)). In addition to the predominant graphite carbon peak at high binding energy of 284.8 eV, commercial WC and W₂C powders show a sharp carbide C 1s signal at a lower binding energy (~282.7 eV) (Supplementary Fig. 3a), but it is absent in W/WO₂ and pure carbon materials, suggesting the graphite carbon species is dominant in the as-prepared W/WO₂ materials. Raman spectra was used to evaluate the quality of graphite carbon in accordance with the D/G ratio of W/WO₂ materials. The D/G ratio is determined to be 0.87, which is even lower than that of the as-prepared bare carbon materials (D/G=0.9) (Supplementary Fig. 3b), indicating the as-prepared W/WO₂ materials are supported by relatively high-quality graphite-carbon substrates

In addition, we would like to point out a small detail in the synthesis of tungsten oxides and tungsten carbides. Usually, the formation of W-C carbide bonds requires high temperature over 750 °C by the carburization of tungsten oxide precursors (*Nat. Mater.* **20**, 1240-1247 (2021); *J. Am. Chem. Soc.* **139**, 5285-5288 (2017)), while tungsten oxides supported and/or encapsulated by carbon materials are obtained at a lower temperature (700 °C) (*J. Am. Chem. Soc.* **137**, 6983-6986 (2015)). In our work, the pyrolyzed temperature is selected to be 700 °C for the synthesis of desired W/WO₂ materials, which provides a reduction condition for the transformation of high-valence W₁₈O₄₉ precursors to low-valence W⁰/W⁴⁺ species, but cannot result in the formation of W-C carbide bonds.

Accordingly, the relevant texts have been added in the revised manuscript.

“Meanwhile, the possibly formed tungsten-carbide by-products (e.g., WC, W₂C) are excluded by X-ray photoelectron spectroscopy (XPS) and Raman characterizations (Supplementary Fig. 3)”

“X-ray photoelectron spectroscopy (XPS) is very sensitive to identify the surface carbon species in carbon supported active materials (*ACS Nano* **9**, 5125-5134 (2015)). In addition to the predominant graphite carbon peak at high binding energy of 284.8 eV, commercial WC and W₂C powders show a sharp carbide C 1s signal at a lower binding energy (~282.7 eV) (Supplementary Fig. 3a), but it is absolutely absent in W/WO₂ and pure carbon materials, suggesting the graphite

carbon species is dominant in the as-prepared W/WO₂ materials. Raman spectra was used to evaluate the quality of graphite carbon in accordance with the D/G ratio of W/WO₂ materials. The D/G ratio is determined to be 0.87, which is even lower than that of the as-prepared bare carbon materials (D/G=0.9) (Supplementary Fig. 3b), indicating the as-prepared W/WO₂ materials are supported by relatively high-quality graphite-carbon substrates.”

Supplementary Figure 3. The identification of surface carbon species in the as-prepared W/WO₂ materials. (a) C 1s XPS and (b) Raman spectra.

Comments 2:

By comparing Figure 4b and Figure S14, it can be observed that both pure-phase W and WO₂ do not have the signals of ads. H₂O. Please explain this phenomenon.

Author reply: Thanks for the referee’s careful reading and constructive suggestion. According to the high-resolution TEM observation, high-density of W nanoparticles are dispersed on WO₂ substrates (Fig. 1a and b), which provides abundant W/WO₂ interfaces for the following water activation in the analysis chamber of NAP-XPS. It is well known that the interfaces with rich unsaturated sites are expected to serve as the ideal adsorbed sites for H₂O molecules (*Nat. Commun.* **9**, 1809 (2018)), as also evidenced by the ESR and O K-edge NEXAFS characterizations (Fig. 2c and Supplementary Fig. 6), which implies that W/WO₂ heterostructure catalyst can provide more available active sites for water adsorption in comparison with W and WO₂ counterparts. On the other hand, the detection sensitivity of NAP-XPS is almost one order of magnitude lower than that of traditional ultrahigh vacuum XPS (UHV-XPS), the relatively lower detection ability may cause the negligence of weak ads. H₂O signal of W and WO₂ materials, meanwhile, the water atmosphere may further inhibits the collection of photoelectron signals during in situ NAP-XPS measurements. Accordingly, the relevant text has been added in the revised Supplementary Information file. “Compared to the distinct ads. H₂O signal of W/WO₂

sample, no relevant signals can be observed on W and WO₂ sample, which can be understood by two reasons: (i) the high-density interfaces of W/WO₂ with rich unsaturated sites are expected to serve as the ideal adsorbed sites for H₂O molecules (*Nat. Commun.* **9**, 1809 (2018)), which implies that W/WO₂ heterostructure catalyst can provide more available active sites for water adsorption in comparison with W and WO₂ counterparts; (ii) it should be noted that the detection sensitivity of NAP-XPS is almost one order of magnitude lower than that of traditional ultrahigh vacuum XPS (UHV-XPS), the relatively lower detection ability may cause the negligence of weak ads. H₂O signal of W and WO₂ materials, meanwhile, the water atmosphere may further inhibits the collection of photoelectron signals from W and WO₂ catalyst surface during in situ NAP-XPS measurements.”

Comments 3:

For Figure 4h, it can be seen that with the increase of applied potential, the proportion of W-OH_T/W-OH_L decreases. Please analyze this phenomenon.

Author reply: Thanks for the referee’s valuable comment. In our work, with the increase of applied potentials, the proportion of W-OH_T/W-OH_L decreases mainly due to the continuously increased lattice hydrogen species (W-OH_L). The increasing negative potentials can result in more electron-rich water molecules being adsorbed on W/WO₂ catalyst surface, and the water molecules rapidly react with electrons for the cleavage of H-OH bonds, then the produced H* intermediates insert into the lattice of tungsten oxides (W-OH_L) for constructing proton-concentrated catalyst surface with electronic and chemical environments approaching to those of H₂WO₄ materials. While the terminated electron-rich hydroxyl species, such as the produced OH* intermediates and chemically/physically adsorbed OH/H₂O molecules (W-OH_T) cannot continuously increase because of the enhanced Coulomb effect at cathode. Accordingly, the relevant text has been added in the revised manuscript. “Meanwhile, we also notice that the proportion of W-OH_T/W-OH_L decreases with the increase of applied potentials, which may be attributed to the continuously increased coverage of produced protons. The increasing negative potentials can result in more water molecules being adsorbed on W/WO₂ catalyst surface, and the water molecules rapidly react with electrons for the cleavage of H-OH bonds, then the produced H* intermediates insert into the lattice of tungsten oxides (W-OH_L) for constructing proton-concentrated catalyst surface with electronic and chemical environments approaching to those of H₂WO₄ materials. While the

terminated electron-rich hydroxyl species, such as the produced OH* intermediates and chemically/physically adsorbed OH⁻ molecules (W-OH_T) cannot follow the increasing tendency because of the enhanced Coulomb effect at cathode.”

Comments 4:

In the DFT calculation of water dissociation, some literatures have reported that water adsorbs on the material surface to produce H₂O*, and the free energy of this step is not 0 eV. Please confirm whether the results in Figure 5b need to be corrected.

Author reply: Thanks for the referee’s valuable comment. The constructive suggestion makes the simulation results more reasonable. We have calculated the adsorption energies of H₂O molecules on the three types of tungsten-based catalyst surfaces in Fig. 5b. WO₂ (-0.33 eV) and W/WO₂ (-0.32 eV) materials exhibit more negative adsorption energies of H₂O molecules than that of W sample (-0.14 eV) (Fig. 5b), which suggests the WO₂ and W/WO₂ catalyst surfaces are beneficial for the adsorption and activation of H₂O reactants (*Sci. Adv.* **7**, eabd6647 (2017); *Nat. Commun.* **13**, 5785 (2022)). Accordingly, the relevant text has been added in the revised manuscript. “WO₂ (-0.33 eV) and W/WO₂ (-0.32 eV) materials exhibit more negative adsorption energies of H₂O molecules than that of W sample (-0.14 eV) (Fig. 5b), which suggests the WO₂ and W/WO₂ catalyst surfaces are beneficial for the adsorption and activation of H₂O reactants (*Sci. Adv.* **7**, eabd6647 (2017); *Nat. Commun.* **13**, 5785 (2022)).”

Fig. 5 (b) The calculated adsorption energies of H₂O molecules on the surface of W, WO₂, and W/WO₂ catalyst, respectively.

Comments 5:

The authors used methods such as REELS, NMR, Py-IR, and NAP-XPS in the experiment. Please supplement detailed test steps in the "Method" section.

Author reply: Thanks for the referee’s valuable comment. According to the suggestion of the

referee, more detailed measurement information have been added in the revised “Method” section.

All relevant texts are listed as follows:

“For NAP-XPS characterization, in situ NAP-XPS measurements were conducted on a SPECS NAP-XPS instrument, where the photon source is the monochromatic X-ray source of Al K α (1486.6 eV), and the overall spectra resolution is Ag 3d $_{5/2}$, < 0.5 eV FWHM at 20 kcps@UHV. The dried sample was transferred to the analysis chamber of XPS, and then W 4f, C 1s, O 1s XPS profiles were simultaneously recorded under the pressure of 1×10^{-9} mbar. After introducing water molecules, the corresponding XPS signals were collected under the water atmosphere with a pressure of 1×10^{-1} mbar.”

“For TOF-SIMS characterization, TOF-SIMS measurement was performed on a TOF-SIMS 5-100 instrument (ION-TOF GmbH) using a 30-keV Bi $^{3+}$ as analysis beam for negative and positive polarity measurements. The analysis beam current, raster size, and incident angles of all beam are 0.7 pA, 50 x 50 μm^2 , and 45°, respectively.”

“For REELS characterization, the electrochemically treated materials were dried, and then transferred to the analysis chamber for REELS characterization using ESCALAB Xi equipment (Thermo Scientific). In detail, approximately 1000 eV electrons were incident on the surface of samples, the XPS analyzer would investigate the elastic and non-elastic scattering, corresponded to the elastic signal and energy loss peak (H loss). In our measurements, all parameters, such as the total acquisition time, energy step size, number of energy steps, and number of scans were set to be 2 mins, 0.02 eV, 1001, and 4, respectively.”

“For NMR characterization, ^1H Magic angle spinning nuclear magnetic resonance (MAS NMR) spectra was collected on a Bruker AVANCE NEO 400 MHz Wide Bore spectrometer operating at a magnetic field of 9.40 T. The chemical shifts for ^1H MAS NMR spectra was referenced to tetramethyl (TMS). ^1H MAS NMR spectra was acquired at a spinning rate of 15 kHz with a $\pi/2$ pulse width of 3.5 μs and s recycle of 5s.”

“For Py-IR characterization, Py-IR measurements were performed on a Nicolet 380 FT-IR instrument (Thermo Co., USA). The catalyst sample was pretreated at 353 K for 2 h under a vacuum condition, and a background spectra was recorded in the ranging of 1700~1400 cm^{-1} before adsorbing pyridine molecules. After reaching the adsorption equilibrium, the catalyst sample was retreated at 353 K for 4 h under a vacuum condition for the complete remove of

physically adsorbed molecules, and the Py-IR spectra of pyridine chemisorption was obtained.”

Replies on comments of reviewer 3:

The authors synthesized a heterostructure composed of tungsten nanoparticles and WO₂ crystals, acting as a hydrogen evolution reaction (HER) catalyst on nickel foam. The catalyst was prepared by calcinating a mixture of W₁₈O₄₉ nanowires and an organic carbon source. For comparative purposes, the authors also prepared bare W and WO₂ catalysts for the HER. The W/WO₂ catalyst demonstrated significantly superior catalytic performance for the HER in alkaline conditions compared to the other two catalysts. Its enhanced activity level was roughly equivalent to that of a noble metal catalyst. Notably, the current density was consistently maintained for 50 hours. This research intriguingly suggests the potential of utilizing low-cost metal oxides as HER catalysts in alkaline conditions. However, the characterization and illustration is not enough to describe the structural properties of W/WO₂ heterostructure, and additional demonstration is required to support the author’s statement. Thus, this manuscript should be revised before publication in the Nature Communications.

Author reply: Thanks for the reviewer’s valuable and constructive comments. The main concern of reviewer 3 is also raised by another two reviewers from different viewpoint, we have supplemented additional experiments and demonstrations to better illustrate the advantage of W/WO₂ heterostructure in alkaline HER process.

Comments 1:

There is a discrepancy between the terminology used within the manuscript and that used in the accompanying figures when referring to the prepared tungsten nanoparticles and WO₂ nanorods. For instance, the term 'phase-pure WO₂' is utilized in the manuscript, whereas 'bare WO₂' appears in the figures (for example, Figure 2b or 2c). For consistency, it is recommended to revise the terminology across both the manuscript and the figures.

Author reply: Thanks for the referee’s valuable comment. According to the suggestion of the referee, all ‘tungsten nanoparticles’ and ‘tungsten dioxide nanorods’ have been abbreviated as ‘W’ and ‘WO₂’, respectively, and the inconsistent usage in both main text and figures has been corrected in the revised manuscript.

Comments 2:

The W/WO₂ catalyst was calcined using a 5% H₂/Ar and at 750 °C, and carbon residue was visible in the TEM images (Figure 1 and S4). Could there be a possibility of tungsten carbide (WC) synthesis? WC is typically studied as another type of HER catalyst. To conclude that improved catalytic performance is originated from the W/WO₂, evidence is needed to exclude the presence of WC. C 1s XPS spectra would help confirm that the influence of any potential WC catalyst on the HER can be disregarded.

Author reply: Thanks for the referee's constructive comments. X-ray photoelectron spectroscopy (XPS) is very sensitive to identify the surface carbon species in carbon supported active materials (*ACS Nano* **9**, 5125-5134 (2015)). In addition to the predominant graphite carbon peak at high binding energy of 284.8 eV, commercial WC and W₂C powders show a sharp carbide C 1s signal at a lower binding energy (~282.7 eV) (Supplementary Fig. 3a), but it is absent in W/WO₂ and pure carbon materials, suggesting the absence of tungsten carbide species in the as-prepared W/WO₂ materials. Raman spectra was used to evaluate the quality of graphite carbon in accordance with the D/G ratio for W/WO₂ materials. The D/G ratio is determined to be 0.87, which is even lower than that of the as-prepared bare carbon materials (D/G=0.9) (Supplementary Fig. 3b), indicating the as-prepared W/WO₂ materials are supported by relatively high-quality graphite-carbon substrates

In addition, we would like to point out a small detail in the synthesis of tungsten oxides and tungsten carbides. Usually, the formation of W-C carbide bonds requires high temperature over 750 °C by the carburization of tungsten oxide precursors (*Nat. Mater.* **20**, 1240-1247 (2021); *J. Am. Chem. Soc.* **139**, 5285-5288 (2017)), while tungsten oxides supported and/or encapsulated by carbon materials are obtained at a lower temperature (700 °C) (*J. Am. Chem. Soc.* **137**, 6983-6986 (2015)). In our work, the pyrolyzed temperature is selected to be 700 °C for the synthesis of desired W/WO₂ materials, which provides a reduction condition for the transformation of high-valence W₁₈O₄₉ precursors to low-valence W⁰/W⁴⁺ species, but cannot result in the formation of W-C carbide bonds.

Accordingly, the relevant texts have been added in the revised manuscript.

“Meanwhile, the possibly formed tungsten-carbide by-products (e.g., WC, W₂C) are excluded by X-ray photoelectron spectroscopy (XPS) and Raman characterizations (Supplementary Fig. 3).”

“X-ray photoelectron spectroscopy (XPS) is very sensitive to identify the surface carbon species in carbon supported active materials (*ACS Nano* **9**, 5125-5134 (2015)). In addition to the predominant graphite carbon peak at high binding energy of 284.8 eV, commercial WC and W₂C powders show a sharp carbide C 1s signal at a lower binding energy (~282.7 eV) (Supplementary Fig. 3a), but it is absolutely absent in W/WO₂ and pure carbon materials, suggesting the absence of tungsten carbide species in the as-prepared W/WO₂ materials. Raman spectra was used to evaluate the quality of graphite carbon in accordance with the D/G ratio for W/WO₂ materials. The D/G ratio is determined to be 0.87, which is even lower than that of the as-prepared bare carbon materials (D/G=0.9) (Supplementary Fig. 3b), indicating the as-prepared W/WO₂ materials are supported by relatively high-quality graphite-carbon substrates.”

Supplementary Figure 3. The identification of surface carbon species in the as-prepared W/WO₂ materials. (a) C 1s XPS and (b) Raman spectra.

Comments 3:

The authors attributed the enhanced catalytic performance of the W/WO₂ heterostructure to the abundance of oxygen vacancies, as revealed by ESR and NEXAFS analyses at the O K edge. However, the XPS O 1s spectra, which also contain the information about the oxygen vacancies, were not presented in this manuscript. XPS results could provide additional support by confirming the quantity of oxygen vacancies formed within the prepared catalyst.

Author reply: Thanks for the referee’s constructive suggestion. Generally, O 1s XPS spectra of the prepared tungsten-based catalysts have been presented in Fig. 4b and Supplementary Fig. 16, and the relative concentrations of oxygen vacancies (O_v) within W, WO₂, and W/WO₂ materials

have been calculated according to the deconvolution of O 1s XPS spectra (Supplementary Table 3). For the O 1s XPS profile of W/WO₂ sample, which exhibits two peaks at 530.3 and 531.3 eV after deconvolution, corresponding to the lattice oxygen (W-O) species and oxygen vacancies (O_v), respectively (*J. Mater. Chem. A*, **7**, 14592-14601 (2019)). W/WO₂ exhibits relative concentration of O_v is as high as 30.5 at%, which is beneficial for the regulation of chemical and electronic structures of neighboring tungsten atoms (*Nat. Commun.* **6**, 8064 (2015)), and is also expected to serve as active sites for the adsorption and activation of water molecules (*Proc. Natl. Acad. Sci. U. S. A.* **120**, e2217148120 (2023)).

Fig. 4 (b) O 1s XPS spectra of W, WO₂, W/WO₂ samples and the corresponding concentrations of O_v according to the XPS deconvolution.

Comments 4:

In Figure S11, a new NEXAFS peak emerges for the used W/WO₂ at 540 eV. Commonly, an O K-edge NEXAFS spectra peak around 540 eV is indicative of interaction between the O 2s orbital and the metal *sp* state. Please provide further explanation about the significance and implications of this new peak.

Author reply: Thanks for the referee's valuable comment. In addition to the change of peak at low photo energy (~532.3 eV), we also observe that the used W/WO₂ sample exhibits a markedly enhanced band at approximately 538.8 eV, such a distinct peak can be attributed to the electronic interactions between O 2*p* and metal *sp* orbitals in traditional 3d metal (Mn, Fe, Co, Ni) oxides (*Angew. Chem. Int. Ed.* **58**, 11720-11725 (2019)), whereas the hybridization (O 2*p*-W 5*d* (e_g)) between O 2*p* and W 5*d* (e_g) orbitals should be responsible for the broad signal at approximately 538.8 eV in non-3d metal oxides (*Ionics* **4**, 101-105 (1998)). Moreover, the sharply increased intensity of O 2*p*-W 5*d* (e_g) hybridization directly suggests partially covalent interaction between oxygen and hydrogen atoms (*Ionics* **4**, 101-105 (1998)).

Accordingly, the relevant discussion has been added in the revised Supplementary Information file. “In addition to the change of peak at low photo energy (~ 532.3 eV), we also observe that the used W/WO₂ sample exhibits a markedly enhanced band at approximately 538.8 eV, such a distinct peak can be attributed to the electronic interactions between O 2p and metal sp orbitals in traditional 3d metal (Mn, Fe, Co, Ni) oxides (*Angew. Chem. Int. Ed.* **58**, 11720-11725 (2019)), whereas the hybridization (O 2p-W 5d (e_g)) between O 2p and W 5d (e_g) orbitals should be responsible for the broad signal at approximately 538.8 eV in non-3d metal oxides (*Ionics* **4**, 101-105 (1998)). Moreover, the sharply increased intensity of O 2p-W 5d (e_g) hybridization directly suggests partially covalent interaction between oxygen and hydrogen atoms (*Ionics* **4**, 101-105 (1998)), confirming the insertion of produced hydrogen atoms into tungsten-oxide lattices after alkaline HER process”

Supplementary Figure 13. O K-edge near edge X-ray absorption fine structure (NEXAFS) spectroscopy of the fresh and used W/WO₂ catalysts.

Comments 5:

The illustration of Figure 4e and f was omitted in the manuscript and caption. Please describe the meaning of that figure.

Author reply: Thanks for the referee’s careful reading and valuable suggestion. The relevant description has been added in the revised manuscript. “In order to visually display the distribution of H₃O⁺ on the electrochemically treated W/WO₂ catalyst surface, two-dimensional (2D) image analysis of the soaked and electrochemically treated W/WO₂ samples are also provided in Fig. 4e and f. Obviously, one could notice that the concentration of H₃O⁺ species on the electrochemically treated W/WO₂ catalyst is much higher than that of only soaked counterpart, indicating the achievement of proton-concentrated catalyst surface on W/WO₂ materials.”

Comments 6:

What about assessing the catalytic activity of the W, WO₂, and W/WO₂ catalysts using the Rotating Disk Electrode (RDE) technique? Evaluating the HER activity of the prepared catalysts through RDE measurements could provide a more quantitative comparison of their catalytic activity relative to previously reported catalysts.

Author reply: Thanks for the referee's valuable comment. In accordance with the constructive suggestion of the referee, the HER performance of W, WO₂, and W/WO₂ catalysts were also examined in 1 M KOH electrolyte using the rotating disk electrode technique (Pine instrument). As can be seen, W/WO₂ catalyst still exhibits a remarkable alkaline HER activity with a low overpotential (-60 mV) at -10 mA/cm² and a small Tafel slope (-54 mV/dec), which are more excellent than those of bare W and WO₂ counterparts, and still excelling most previously reported metal oxides.

Accordingly, the relevant texts have been added in the revised manuscript.

“Moreover, the HER performance of W, WO₂, W/WO₂ powders were also examined using the rotating disk electrode (RDE) technique in 1 M KOH electrolyte. As can be seen, W/WO₂ catalyst still exhibits a remarkable alkaline HER activity with a low overpotential (-60 mV) at 10 mA/cm² and a small Tafel slope (-54 mV/dec), which are more excellent than those of W and WO₂ counterparts, and still excelling most previously reported metal oxides (Fig. 3c).”

“For the preparation of W, WO₂, W/WO₂ catalyst ink, 5 mg of the powder materials is dispersed in the mixture of 800 uL ethanol, 170 uL water, and 30 uL nafion, followed by sonication treatment at least 30 min for the preparation of homogeneous catalyst ink. Then, 20 uL of the catalyst ink was dropped on the polished glassy carbon rotating disk electrode for electrocatalytic RDE measurements at 1600 rpm.”

Supplementary Figure 8. The evaluation of HER performance of W, WO₂, and W/WO₂ catalysts

in 1 M KOH electrolyte using the rotating disk electrode technique at 1600 rpm. (a) Polarization curves and (b) Tafel plots of W, WO₂, and W/WO₂ catalysts.

Comments 7:

The authors computed the transition energy for the water dissociation pathway on W, WO₂, and at the W/WO₂ interface, with the corresponding simulated models displayed in Figures 5 and S18. Could the authors specify the bond length between the oxygen and dissociated hydrogen atom after dissociation? It appears that this bond length on the W/WO₂ interface may be shorter than in the other two cases, which might contribute to the smaller energy barrier. Additionally, more comprehensive information, such as top and side images of the calculated structure, would be beneficial to better illustrate the W/WO₂ interface.

Author reply: Thanks for the referee's valuable comments. We feel sorry for not describing the H₂O dissociation process clearly. The bond lengths between oxygen and dissociated hydrogen atom (O··H) on W, WO₂, and W/WO₂ catalyst surface are 2.98, 2.84, and 2.75 Å, respectively (Supplementary Table 5). Generally, the differences in O··H bond lengths on above three-types tungsten-based catalyst surfaces are very small, and the O··H bond length of W/WO₂ interface appears much shorter than the other two cases mainly due to the drawing perspective. Therefore, the major contribution of low activation barrier of H₂O molecules is the regulated chemical and electronic structures rather than the shorter dissociated length of O··H on W/WO₂ interface. In addition, we agree with the referee that more comprehensive information about the calculated structures should be provided, and we supplemented with additional front, side, and top images of the calculated structures to make the substrate, adsorbed H₂O and dissociated H₂O on W/WO₂ interface clearer in the revised manuscript (Supplementary Fig. 21).

Accordingly, the relevant discussion has been added in the revised Supplementary Information file. “The bond lengths between oxygen and dissociated hydrogen atom (O··H) on W, WO₂, and W/WO₂ catalyst surface are 2.98, 2.84, and 2.75 Å, respectively (Supplementary Table 5). Generally, the differences in O··H bond lengths on above three-types tungsten-based catalyst surfaces are very small, and the O··H bond length of W/WO₂ interface appears much shorter than the other two cases mainly due to the drawing perspective. Therefore, the major contribution of low activation barrier of H₂O molecules is the regulated chemical and electronic structures rather than the shorter dissociated length of O··H on W/WO₂ interface.”

”

Supplementary Figure 21. Front, side and top images of the calculated structure for W/WO₂ interface. H₂O molecule undergoes water adsorption, activated H₂O adsorption, produced H and OH adsorption on W/WO₂ interface in alkaline HER process.

Supplementary Table 5. The bond lengths between oxygen and hydrogen atom within H₂O molecule before and after dissociation on W, WO₂, W/WO₂ catalyst surface.

State	Bond length between O and dissociated H (Å)		
	W	WO ₂	W/WO ₂
Before	0.98	1.02	1.02
After	2.98	2.84	2.75

Comments 8:

The author insisted that heterojunction between the W nanoparticle and WO₂ structure is the reason of the improved catalytic activity. How about the HER activity of the catalyst that synthesized by physical mixing of the W nanoparticle and WO₂ nanorod?

Author reply: Thanks for the referee’s valuable comment. In accordance with the constructive suggestion of the referee, the HER activity of physical mixture of W and WO₂ powders (W+WO₂) has been evaluated in alkaline electrolyte. As can be seen, the alkaline HER activity of W+WO₂ is still inferior to W/WO₂ catalyst, suggesting the W/WO₂ interfaces constructed by chemical bonds can indeed improve the alkaline HER activity intrinsically.

Accordingly, the following texts have been added in the revised manuscript.

“In addition, the alkaline HER activity of the physical mixture of W and WO₂ (W+WO₂) was

also examined under the same conditions, where the values of η_{10} and Tafel slope are determined to be -153 mV and -135 mV/dec, respectively, and the markedly improved Tafel slope may be attributed to the synergistic effect of WO_2 and W components for water dissociation and hydrogen desorption steps in alkaline HER process. However, the alkaline HER activity of $\text{W}+\text{WO}_2$ is still inferior to W/WO_2 catalyst, suggesting the W/WO_2 interfaces constructed by chemical bonds can improve the alkaline HER activity intrinsically.”

“For the fabrication of free-standing electrode for the physical mixture of W and WO_2 powders ($3.2 \text{ mg}/\text{cm}^2$), W (5 mg) and WO_2 (5 mg) were dispersed into in a mixture of 800 μL ethanol, 170 μL water, and 30 μL nafion, followed by sonication treatment over 30 min for preparation of the homogeneous catalyst ink. Then, 200 μL of the mixture catalyst ink was dropped onto the pre-treated Ni and pressed into a thinner foil for the following electrocatalytic measurement.”

Fig. 3 (a) Polarization (LSV) curves and (b) Tafel plots of C@Ni, W, WO_2 , $\text{W}+\text{WO}_2$, W/WO_2 and commercial PtRu@C catalysts.

REVIEWERS' COMMENTS

Reviewer #1 (Remarks to the Author):

The Authors have addressed all the comments appropriately and significantly improved their manuscript. Considering the importance of the findings reported, I believe that the manuscript can now be published without any additional changes.

Reviewer #2 (Remarks to the Author):

The authors have well addressed all my concerns and I would like to suggest its acceptance as it is now.

Reviewer #3 (Remarks to the Author):

I am writing to express my satisfaction with the revised manuscript. Their thoughtful and responsive approach to the feedback given has significantly enhanced the clarity and quality of the research.

It is evident that the author has diligently incorporated the feedback, leading to a marked improvement in the manuscript. Notably, they have provided extra experimental evidence and additional characterization, yielding a more in-depth understanding of the reaction system under study. These updates not only strengthen their arguments but also substantiate their conclusions.

Moreover, the manuscript has greatly benefitted from the revision in terms of readability. The author has made commendable efforts in rectifying ambiguous sections and descriptions, making the content more comprehensible for the readers. This certainly elevates the quality of the manuscript, aligning it with the high standards expected by Nature Communications.

Given these substantial improvements, I am confident in my decision to support the manuscript's publication in Nature Communications. I extend my appreciation to the author for their diligent and thorough response to the feedback.

Answer to the comments of reviewers:

Replies on comments of reviewer 1:

The Authors have addressed all the comments appropriately and significantly improved their manuscript. Considering the importance of the findings reported, I believe that the manuscript can now be published without any additional changes.

Author reply: we appreciate the reviewer's recommendation very much. We are delighted that our revised version of the manuscript satisfies the reviewer. We would like to thank again the reviewer's constructive comments and suggestions for substantially improving the quality of our manuscript.

Replies on comments of reviewer 2:

The authors have well addressed all my concerns and I would like to suggest its acceptance as it is now.

Author reply: We appreciate the reviewer's recommendation very much.

Replies on comments of reviewer 3:

I am writing to express my satisfaction with the revised manuscript. Their thoughtful and responsive approach to the feedback given has significantly enhanced the clarity and quality of the research.

It is evident that the author has diligently incorporated the feedback, leading to a marked improvement in the manuscript. Notably, they have provided extra experimental evidence and additional characterization, yielding a more in-depth understanding of the reaction system under study. These updates not only strengthen their arguments but also substantiate their conclusions.

Moreover, the manuscript has greatly benefitted from the revision in terms of readability. The author has made commendable efforts in rectifying ambiguous sections and descriptions, making the content more comprehensible for the readers. This certainly elevates the quality of the manuscript, aligning it with the high standards expected by Nature Communications.

Author reply: We would like to thank again the reviewer's acknowledgement that our added experimental evidence and additional characterization yield a more in-depth understanding of the

reaction system under study, and we also appreciate the reviewer's recommendation of this manuscript to be published in Nature Communications.